# From Drift to Coherence: Stabilizing Beliefs in LLMs

**SongEun Kim** [1]  **Seungyoo Lee** [2]  **Edwin Fong** [3]  **Hyungi Lee**[† 4]  **Juho Lee**[† 2]

## Abstract

Large language models (LLMs) are often hypothesized to perform implicit Bayesian inference, yet a key coherence condition—the martingale property of predictive beliefs—has been shown to fail in controlled synthetic in-context learning settings. We revisit this question in a more typical usage regime: generic multiple-choice question answering. Exploiting the discrete answer space, we compute exact predictive distributions and study belief dynamics induced by autoregressive answer resampling. We introduce prompted predictive resampling (PPR), where an LLM generates a sequence of answers to the same question. Empirically, PPR reveals early-stage belief drift, indicating martingale violations. However, after sufficient resampling steps, the belief process self-stabilizes and converges to a coherent predictive distribution. Based on this observation, we further propose (i) a seed-answer prompting strategy to accelerate stabilization, and (ii) a self-consistency loss that amortizes early-stage drift into the model via fine-tuning. Experiments on multiple-choice QA benchmarks show that our methods substantially reduce belief drift and improve predictive coherence without sacrificing accuracy.

## 1. Introduction

Among the various capabilities of large language models (LLMs; Brown et al., 2020; Guo et al., 2025; Singh et al., 2025; Comanici et al., 2025), a particularly striking one is their ability to rapidly adapt or generalize from only a small number of examples across a wide range of tasks. A canonical manifestation of this behavior is in-context learning (ICL), in which an LLM is prompted with a sequence of example input–output pairs and then asked to produce an output for a new input, without any parameter updates or explicit fine-tuning (Brown et al., 2020; Dong et al., 2024), although they were not explicitly trained to do that.

There has been a growing body of work seeking to explain ICL, and more broadly the rapid adaptability of large language models, through the lens of Bayesian inference (Xie et al., 2022; Jiang, 2023; Wang et al., 2024; Ye et al., 2024; Falck et al., 2024). A common line of reasoning in this literature is to demonstrate—either in analytically simplified settings or via empirical analyses—that LLMs implicitly infer a latent concept or knowledge variable $\theta$ that summarizes the in-context examples, and subsequently generate predictions for new queries by conditioning on $\theta$ (Xie et al., 2022; Jiang, 2023; Wang et al., 2024). If true, this perspective would render LLMs as Bayesian predictives arising from an implicitly defined prior–likelihood pair over the latent variable $\theta$.

The question of whether LLMs perform implicit Bayesian inference is part of a more general inquiry: namely, whether autoregressive generative models—or more broadly, *one-step predictive sequences*—trained without an explicit latent variable $\theta$ can nevertheless be associated with such a $\theta$, at least implicitly. This question is closely related to the *predictive modelling paradigm* in Bayesian statistics (Fong et al., 2023; Fortini & Petrone, 2025), which seeks to formulate Bayesian inference purely in terms of predictive sequences. According to this theory, if a predictive sequence is exchangeable, one can prove the existence of an underlying latent variable $\theta$ equipped with a proper prior–likelihood pair. Moreover, if the predictive sequence is *conditionally identically distributed (c.i.d.)* (Berti et al., 2004), or equivalently satisfies a martingale property (Fong et al., 2023), then there exists a latent variable $\theta$ that captures epistemic uncertainty through the so-called *martingale posterior*, which generalizes the classical Bayesian posterior. Leveraging this theoretical connection, Falck et al. (2024) empirically demonstrated violations of the martingale property in LLMs on several synthetic and toy tasks, and consequently concluded that LLMs cannot be regarded as Bayesian in general. However, their analysis is primarily illustrative and is therefore restricted to relatively artificial tasks (e.g., random number generation), which limits its direct relevance to the practical tasks for which LLMs are typically deployed.

---

[†]Equal Correspondence [1]Department of Statistics, Seoul National University [2]Korea Advanced Institute of Science & Technology [3]University of Hong Kong [4]Department of AI, Kookmin University. Correspondence to: Hyungi Lee <lhk2708@kookmin.ac.kr>, Juho Lee <juholee@kaist.ac.kr>.

*Proceedings of the 43$^{rd}$ International Conference on Machine Learning*, Seoul, South Korea. PMLR 306, 2026. Copyright 2026 by the author(s).

In this paper, we extend the martingale perspective on LLMs introduced by Falck et al. (2024) to a broader class of tasks—going beyond ICL and synthetic settings tailored for theoretical alignment—and study the most generic, and arguably most common, mode of LLM usage: responding to a query $Q$ or a prompt centered around it. To this end, we introduce a *prompted predictive resampling* scheme, in which an LLM is asked to answer the same question multiple times *in sequence*, rather than as independent samples. At first glance, this procedure may appear unnatural or even uninformative, as it departs from standard LLM usage and may be perceived as an artificial attempt to force LLM responses into predictive-resampling from martingale posteriors (Fong et al., 2023).

Empirically, however, we observe that while the probabilities over possible answers computed by an LLM may drift in the short run, they stabilize after a sufficient number of steps into a coherent sequence of probabilities. This stabilization effectively renders the LLM a coherent predictive machine equipped with a martingale posterior. We further propose (i) a simple prompting-based trick to accelerate this stabilization and (ii) a fine-tuning strategy that amortizes the required number of resampling steps, derived from a loss enforcing the c.i.d. condition. Experiments on standard question-answering benchmarks show that, once stabilized—either via predictive resampling or amortization—the resulting predictive distributions exhibit improved uncertainty calibration without sacrificing accuracy.

## 2. Preliminaries

### 2.1. Problem Formulation

A growing body of recent work (Xie et al., 2022; Jiang, 2023; Falck et al., 2024; Chlon et al., 2025) has sought to interpret LLMs as approximate Bayesian learners, particularly through the lens of ICL. In this line of research, a central question is whether the predictive behavior of an LLM can be understood as coherent Bayesian updating under a latent prior. A prominent recent analysis by Falck et al. (2024) formalizes this question via the martingale property, a necessary condition for Bayesian learning under exchangeable data, and shows that modern LLMs violate it in controlled synthetic experiments with Bernoulli and Normal posteriors.

While such synthetic tasks with continuous priors enable a direct comparison against a theoretical Bayesian oracle, they introduce a limitation for our purposes: they obscure the model's actual predictive distribution. In particular, when predictions are accessed only through samples, one cannot directly observe or manipulate the true parameterized predictive density, making it difficult to reason about internal belief dynamics rather than sampling artifacts.

In contrast, we exploit the fact that LLMs are fundamentally discrete autoregressive density estimators defined over a finite vocabulary. We therefore restrict the output space to a finite, semantically meaningful answer set $\mathcal{A}$ (e.g., multiple-choice options). This "native" discretization is not merely a modeling convenience, but provides two key advantages that are essential for our analysis.

1. **Direct access to internal beliefs.** The model's predictive distribution over $\mathcal{A}$ can be computed exactly from logits computed the model, without relying on sampling-based approximations or kernel density estimation. This allows us to work directly with the model's internal belief state.

2. **A self-consistency criterion.** Our focus is not on whether the model recovers an external ground-truth likelihood, which may not even be well-defined for generic question answering, but on whether the model's own predictive process is internally coherent over time. In particular, we can directly assess whether successive predictive distributions satisfy a martingale property with respect to the model's own generated history.

Building on this discrete formulation, let $p_\phi$ denote a language model parameterized by static weights $\phi$. For a fixed query $Q$, we model answer generation not as a single-shot prediction, but as a stochastic process producing *a sequence of answers* $A_{1:n}$, autoregressively sampled from the finite answer space $\mathcal{A}$.

To account for different prompting and steering strategies, ranging from standard querying to the seed-based interventions studied later, we introduce the notion of an *initial context* $\mathbf{x}_Q$. The context $\mathbf{x}_Q$ represents the specific realization of the prompt provided to the model, which may consist solely of the query $Q$ or an augmented prompt including additional prefixes.

For a fixed context $\mathbf{x}_Q$, the model induces a history-dependent predictive distribution, which we parameterize by

$$\theta_n(\mathbf{x}_Q) := p_\phi(A_{n+1} = \cdot \mid A_{1:n}, \mathbf{x}_Q) \in \mathcal{P}(\mathcal{A}). \quad (1)$$

We take $\theta_n$ as our primary object of interest. Unlike the static model parameters $\phi$, $\theta_n$ is a random variable whose evolution is driven by the model's *own generated answers*. As $n$ increases, $\theta_n$ traces a stochastic trajectory in the probability simplex $\mathcal{P}(\mathcal{A})$, capturing how the model's internal beliefs evolve as it conditions on its own outputs.

This perspective allows us to study Bayesian coherence at the level of predictive belief dynamics, rather than at the level of external performance or calibration. In the following section, we formalize this intuition by introducing

martingale conditions on the sequence $\{\theta_n\}_{n \geq 1}$ and relating them to Bayesian learning principles.

Throughout the paper, conditioning on $A_{1:n}$ is understood as conditioning on the $\sigma$-field generated by the answer history. In particular, the predictive process $\{\theta_n\}_{n \in \mathbb{N}}$ is an adapted process on filtration $\{\sigma(A_{1:n})\}_{n \in \mathbb{N}}$.

## 2.2. The Martingale Property and Posterior

Our framework builds upon the martingale posterior perspective (Fong et al., 2023) to quantify uncertainty without relying on the exchangeability assumption. Following the definition of *the martingale property* in (Fong et al., 2023; Falck et al., 2024), we condition exclusively on the sequence of answers generated by the language model itself to fit the autoregressive nature of language models strictly.

**Definition 2.1** (The Martingale Property). The predictive process $\{\theta_n(\mathbf{x}_Q)\}_{n \in \mathbb{N}}$ satisfies the martingale property if it forms a vector-valued martingale with respect to the filtration $\mathcal{F}_n = \sigma(\mathbf{x}_Q, A_{1:n})$. That is, for all $n, k \in \mathbb{N}$:

$$\mathbb{E}[\theta_{n+k}(\mathbf{x}_Q) \mid \mathcal{F}_n] = \theta_n(\mathbf{x}_Q) \quad \text{a.s.} \tag{2}$$

This definition equivalently implies that the generated sequence $A_{1:\infty}$ is *conditionally identically distributed* (c.i.d.; Berti et al., 2004), meaning the marginal predictive distribution is invariant to the timing of the future observations (see Proposition B.1 for the proof). Intuitively, this asserts that the model's current belief $\theta_n$ is stable and does not shift systematically.

Beyond ensuring instantaneous stability, the martingale property rigorously guarantees the existence of a valid limiting distribution. Since the belief process $\{\theta_n(\mathbf{x}_Q)\}_{n \in \mathbb{N}}$ is bounded within the probability simplex $\mathcal{P}(\mathcal{A})$, Doob's Theorem (Doob, 1949) ensures that $\theta_n(\mathbf{x}_Q)$ converges almost surely to a limiting random variable $\theta_\infty(\mathbf{x}_Q)$ as $n \to \infty$.

This limit $\theta_\infty$ represents the model's "final" epistemic belief state. We therefore define **the martingale posterior** as the probability distribution of this limit, conditioned on the initial context:

$$\Pi_\infty(\cdot \mid \mathbf{x}_Q) := \text{Law}(\theta_\infty \mid \mathbf{x}_Q), \tag{3}$$

where $\text{Law}(\cdot \mid \mathbf{x}_Q)$ denotes the induced conditional probability measure on the simplex $\mathcal{P}(\mathcal{A})$. By guaranteeing the existence of this limit, the martingale property allows us to transform the evolving autoregressive sequence into a coherent, static probability measure over the simplex, enabling direct uncertainty quantification.

## 3. Empirical Observations

While Falck et al. (2024) has largely focused on synthetic and controlled settings with simple Bernoulli and Normal

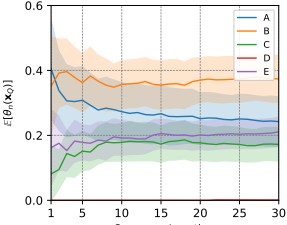 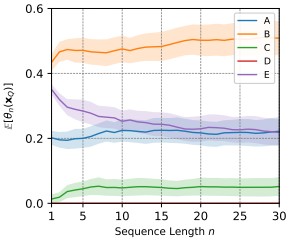

*(a)* Without seed answers     *(b)* With seed answers

*Figure 1.* **Analysis of martingale property violation in prompted predictive resampling.** We instructed LLAMA-3.1-8B to generate a sequence of answers on the CSQA question. Solid lines indicate the mean across independent sample paths, and shaded regions represent $\pm 1$ standard deviation. **(a)** A significant drift during the early generated sequences was observed, indicating the martingale violation. **(b)** Conditioning on a prefix of 5 random i.i.d. sequences effectively mitigates the drift.

distributions, it remains unclear to what extent this property manifests in practical large-scale reasoning tasks encountered by modern LLMs. To address this gap, we evaluate the martingale behavior of the instruction-tuned LLM, LLAMA-3.1-8B (Grattafiori et al., 2024), on a real-world commonsense reasoning benchmark, COMMONSENSEQA (CSQA; Talmor et al., 2019). In this setting, we observe two prominent phenomena: (i) when the predictive distribution is conditioned solely on the question $Q$, the model exhibits a pronounced initial drift in its marginal predictive beliefs, (ii) when the conditioning context is augmented with a short sequence of *seed answers* drawn independent and identically distributed (i.i.d.) from the model without predictive resampling (i.e., the usual way to get answer from the model), this early-stage drift is effectively suppressed.

**Initial Drift as a Violation of Necessary Conditions.** We first examine the marginal distribution of the $n$th answer token generated under predictive resampling. To this end, we augment the query $Q$ with an explicit instruction that allows the model to generate answers repeatedly; the corresponding prompting strategy is detailed in Section 4. Let $\mathbf{x}_Q$ denote the resulting prompt. The marginal distribution of $A_{n+1}$, denoted with our quantity of interest $\theta_n(\mathbf{x}_Q)$, is given by

$$\begin{aligned} \mathbb{E}[\theta_n(\mathbf{x}_Q)] &:= \mathbb{E}_{A_{1:n} \sim p_\phi}[p_\phi(A_{n+1} = \cdot \mid A_{1:n}, \mathbf{x}_Q)] \\ &= p_\phi(A_{n+1} = \cdot \mid \mathbf{x}_Q). \end{aligned} \tag{4}$$

In Figure 1a, we visualize the evolution of these marginal distributions across sampling steps. A pronounced distributional shift is evident during the initial phase, approximately for $n \leq 5$, after which the marginals begin to stabilize. This behavior is particularly informative: stabilization of the marginal expectation is a necessary condition for the martingale property (Definition 2.1). If the average predictive distribution drifts over time, the sequence cannot be conditionally identically distributed. Consequently, the observed

early-stage drift constitutes direct evidence of martingale property violation in the language model during the initial sampling steps.

**Self-Stabilization.** However, Figure 1a also reveals that this distribution *spontaneously stabilizes as the sample length increases*, analogous to the burn-in phase of MCMC chains. While this marginal stabilization is not sufficient to prove the model has become a martingale, it indicates that the process eventually satisfies the minimum distributional requirement for one. This spontaneous equilibration suggests that, specifically within the context of generic query-answering tasks, the violation of the martingale property is largely transient.

**Answer seeding for better stabilization.** Motivated by the observed early-stage drift, we investigate strategies to mitigate violations of the martingale property during the initial sampling steps. We find that supplying the model with additional contextual information can steer the generative process into a stable regime more rapidly. Concretely, we first generate a set of *seed answers*, sampled i.i.d. from $p_\phi(\cdot \mid Q)$; that is, these answers are produced by a standard LLM invocation without any instruction for repeated answering. We then augment the prompt $\mathbf{x}_Q$ by appending both the predictive-resampling instruction and the seed answers. We refer to this strategy as *answer seeding*. Empirically, answer seeding substantially reduces the early-stage distributional drift, leading to faster stabilization of the resulting *seed-marginalized* predictive distribution, as illustrated in Figure 1b.

Taken together, these observations indicate that, in practical question-answering settings, the predictive distribution of an LLM admits regimes in which martingale consistent behavior can be attained. Further details on the experimental setup (e.g., prompt) are provided in Appendix D.

**Intuition on why self-stabilization happens.** Although the mechanism behind stabilization is not fully transparent, a potential high-level intuition is that predictive resampling creates a feedback loop in which the model conditions on its own recent answers. Concretely, the sequence of past answers provides additional context about the model's current uncertainty over the candidate set: if the model's predictive distribution is already sharply concentrated, resampling tends to reproduce the same answer, whereas if multiple answers remain plausible, the resampled sequence will exhibit diversity. Conditioning on this realized history effectively supplies the model with a short summary of its own predictive variability, which can reduce transient fluctuations in the next-step predictive distribution. In this view, the apparent "burn-in" corresponds to the initial stage where only a few past answers are available and the induced context is

too weak to constrain the predictive distribution; once sufficiently many answers have accumulated, the feedback becomes informative enough that the model's predictive probabilities stabilize and become approximately self-consistent across subsequent resampling steps.

**Intuition on answer seeding.** The steering effect of seed answers can be interpreted as follows. Suppose that, under the plain prompt $Q$ (i.e., without explicitly instructing predictive resampling), the model's answer sequence is c.i.d. with one-step predictive $p_\phi(A_{n+1} \mid Q)$. Then the seed answers $S_1, \ldots, S_m \overset{\text{i.i.d.}}{\sim} p_\phi(A_{n+1} \mid Q)$ follow the same marginal distribution as each element of the subsequent resampling trajectory $(A_{n+1}, A_{n+2}, \ldots)$ that would be generated from $p_\phi(\cdot \mid Q)$ (ignoring the additional resampling instruction needed). In this sense, the seeds provide a rough preview of what the model would likely produce in the next few steps: they approximate the expected empirical frequencies of future answers, while implicitly treating those future draws as independent and thus ignoring any correlations within $(A_{n+1}, \ldots, A_{n+m})$.

Viewed this way, appending the seed answers to the prompt supplies the model with a crude "mean-field" summary of its own predictive uncertainty early on, which can reduce the amount of resampling needed before the predictive probabilities stabilize. Crucially, because these seeds are generated endogenously from the model's own distribution, this stochastic self-seeding provides a highly efficient route to a stabilized regime that remains firmly anchored to the model's original predictive behavior, rather than being driven toward an arbitrary exogenous context.

## 4. Predictive Resampling and Amortization

Based on the observations in Section 3, we find that the predictive distribution of an LLM admits regimes in which martingale-consistent behavior can be attained, despite exhibiting transient violations during the early stages of predictive resampling. In this section, we first describe *Prompted Predictive Resampling* (PPR), a prompting strategy that induces self-stabilizing behavior in answer sequences, together with the answer-seeding mechanism introduced above. We then propose a post-hoc tuning method that amortizes the sampling steps required for stabilization by optimizing a loss that explicitly enforces the conditional identically distributed (c.i.d.) property.

### 4.1. Prompted Predictive Resampling

**Resampling protocol into prompt.** To implement the predictive resampling for generic question answering tasks using LLMs, we require a generation protocol that encourages the model to externalize its internal uncertainty as a

concrete sequence of answer tokens. This necessitates not only repeated sampling but also a structured output format that allows individual samples to be unambiguously identified and parsed.

Without such a structure, ambiguities can arise in autoregressive generation. For example, a sequence of identical answers (e.g., "AA") may be produced either as two consecutive tokens "A" "A" or as a single token "AA", depending on the tokenizer and generation context. To eliminate this ambiguity, we explicitly instruct the model to generate exactly one answer token at a time, followed by a newline character ($\backslash$n) as a deterministic delimiter between successive samples.

To ensure strict adherence to this output format and to prevent artificially structured outputs (e.g., cyclic or templated patterns), we include syntactic examples (e.g., `...output might be : C\nA\nC\nC\nA\nA\n ...`) within the prompt. Crucially, these examples serve purely as structural guidance rather than semantic demonstrations. Our analysis confirms that the models do not interpret this setup as a standard few-shot learning scenario; instead, they use the examples solely to infer the required delimiter and layout, without conditioning their reasoning on the example content.

Now, let $I$ be a textual instruction of the generation protocol specified above. We form a prompt $\mathbf{x}_Q$ by prepending $I$ to $Q$, i.e., $\mathbf{x}_Q := [I; Q]$. PPR is then executed by generating answer sequences from $p_\phi(\cdot \mid \mathbf{x}_Q)$. The full prompt specification used in our experiments and examples of generated answer sequences are provided in Appendix D.

**Answer seeding.** In answer seeding, in addition to the instruction $I$, we generate a predefined number of seed answers $S_{1:m} = (S_1, \ldots, S_m) \overset{\text{i.i.d.}}{\sim} p_\phi(\cdot \mid Q)$, drawn i.i.d. from the standard LLM predictive distribution. We then construct the prompt as $\tilde{\mathbf{x}}_Q^{(j)} = [I; Q; S_{1:m}^{(j)}]$. While this construction renders the prompt $\tilde{\mathbf{x}}_Q$ a random variable through its dependence on $S_{1:m}$, the seeded estimator ultimately averages over both rollout randomness and seed randomness. Therefore, it is best interpreted as a seed-marginalized quantity rather than one strictly conditioned on a single fixed seed prefix, allowing us to conceptually treat the overarching marginal process as deterministic conditional on $Q$.

**Recovering target of interest.** In our setup, we have direct access to the predictive probabilities, as $p_\phi(A_{n+1} \mid A_{1:n}, \mathbf{x}_Q)$ are simply computed from LLMs. Still, as we discussed earlier, we may choose a statistic or quantity of interest to be recovered from the sequence $A_{1:N}$. As in Fong et al. (2023), one can recover such a quantity by solving an optimization problem,

$$\theta(A_{1:N}) := \arg\min_\theta \int_{\mathcal{A}} \ell(a, \theta) dF_N(a), \quad (5)$$

where $\ell(a, \theta)$ is a loss function and $F(a)$ is the empirical distribution constructed with the sequence $A_{1:N}$. In our setup, we simply choose $\ell(a, \theta)$ to be the categorical log-likelihood, and the quantity of interest boils down to the Maximum-Likelihood Estimator (MLE),

$$\bar{\theta}_N(A_{1:N} \mid \mathbf{x}_Q) := \arg\max_{\theta \in \mathcal{P}(\mathcal{A})} \sum_{i=1}^{N} \log \text{Cat}(A_i \mid \theta). \quad (6)$$

In case of categorical likelihood, the MLE is simply computed as an empirical frequency vector in the sequence $A_{1:N}$. As $N \to \infty$, the random variable $\bar{\theta}_N(A_{1:N} \mid \mathbf{x}_Q)$ converges almost surely to the model's inherent belief $\theta_\infty$ distributed according to the martingale posterior $\Pi_\infty$ (see Theorem B.2 for the statement and proof). In practice, to simulate the martingale posterior over $\bar{\theta}_N(A_{1:N} \mid \mathbf{x}_Q)$, we draw $J$ independent answer sequences $A_{1:N}^{(j)}$ via PPR for $j = 1, \ldots, J$. As a single representative, we use those trajectories to compute a Monte-Carlo estimator of the posterior mean.

$$q^*(\mathbf{x}_Q) := \mathbb{E}[\theta_\infty \mid \mathbf{x}_Q] \approx \frac{1}{J} \sum_{j=1}^{J} \bar{\theta}_N(A_{1:N}^{(j)} \mid \mathbf{x}_Q). \quad (7)$$

This provides a tractable representative of the limiting predictive belief and enables direct evaluation of martingale consistency. Additionally, it coincides with the Bayes-optimal action under standard proper scoring rules and squared-error loss (Brier Score (Brier, 1950)). Formally, for such seeded prompts $\tilde{\mathbf{x}}_Q$, the law of total expectation guarantees that the expected value of the seed-conditional belief remains perfectly centered on the original unseeded target context $\mathbf{x}_Q = [I; Q]$, satisfying

$$\mathbb{E}_S [\mathbb{E}[\theta_\infty \mid \tilde{\mathbf{x}}_Q]] = \mathbb{E}[\theta_\infty \mid \mathbf{x}_Q]. \quad (8)$$

This mathematically solidifies our interpretation that stochastic self-seeding does not distort the core limiting identity of the model, but rather optimizes the trajectory toward a seed-marginalized family of context-conditional beliefs.

### 4.2. Martingale Amortization

Our empirical analysis in Section 3 revealed that the belief process typically settles into a stable equilibrium as the sequence length grows ($N \to \infty$), regardless of the initial drift. This suggests that the model possesses a coherent *true* belief state, but fails to access it immediately at the start of generation. We call this period of initial instability as a **burn-in phase**, analogous to MCMC. Currently, accessing

the model's true belief requires paying a high computational cost at inference time: either by generating long sequences to wait for convergence or by sampling auxiliary seeds for stochastic steering. Furthermore, while steering effectively stabilizes the process, it introduces external tokens that may subtly shift the semantics of the query.

To address these limitations, we propose *Martingale Amortization*. The core idea is to amortize the burn-in phase directly into the model's parameters. By teaching the model to predict its own future stability, we aim to collapse the convergence horizon, enabling the model to output its internal belief distribution zero-shot without requiring long rollouts or external steering context.

We operationalize this by introducing a supervised fine-tuning loss objective in Definition 4.1 called *Self-Consistency (SC) loss*.

**Definition 4.1** (Self-consistency loss). Let $p_k$ be a $k$-step marginal predictive distribution, defined as $p_{\phi,n}^{(k)} := p_\phi(A_{n+k} \mid a_{1:n}, Q)$, and $\hat{p}_{\phi,n}^{(k)}$ be its estimator obtained from predictive resampling. Let

$$\mathcal{L}_{\text{SC}}(\phi) = \mathbb{E}_{k_1 < k_2}\left[ H(\texttt{stopgrad}(\hat{p}_{\phi,n}^{(k_2)}), \hat{p}_{\phi,n}^{(k_1)}) \right]$$

where $H(p, q)$ is the cross-entropy of $q$ relative to $p$, $\texttt{stopgrad}(\cdot)$ denotes the stop-gradient operator, which treats its argument as a constant during backpropagation, and the expectation is over some prespecified distribution over the pairs of integers $k_1 < k_2$. Since the exact future marginal $p_{\phi,n}^{(k)}$ involves an intractable integral over intermediate paths, we approximate it via Monte Carlo integration as follows:

$$\hat{p}_{\phi,n}^{(k)}(\cdot) := \frac{1}{J} \sum_{j=1}^{J} p_\phi(\cdot \mid A_{n+1:n+k-1}^{(j)}, a_{1:n}, Q).$$

In practice, as discussed earlier, we approximate the expectation over horizon pairs $(k_1, k_2)$ in a truncated fashion: we sample $k_1$ uniformly from $\{1, \ldots, 5\}$ and then sample an offset $\delta := k_2 - k_1$ uniformly from $\{1, \ldots, 6 - k_1\}$, setting $k_2 = k_1 + \delta$. By treating the stabilized distributions in the last stages as our pseudo-ground-truth target, we minimize the divergence between the tail distribution and head distribution. This objective forces the early tokens to match the statistics of the late tokens, effectively "pulling back" the future stability to the present.

Our theoretical analysis confirms that this SC loss is a good proxy for the c.i.d. condition, and hence the martingale condition. We establish that minimizing $\mathcal{L}_{\text{SC}}$ ensures that the predictive process for a fixed initial prompt $\mathbf{x}_Q = Q$ approaches a martingale. In other words, by training the model to match its own future marginal statistics, we recover the

stable belief dynamics of the steered process without requiring auxiliary tokens at inference time. Our main theoretical result formalizes this intuition by assuming the following conditions on the training dynamics:

(A1) $\hat{p}_{\phi,n}^{(k)}$ is uniformly consistent at $p_{\phi,n}^{(k)}$ over $n, k$.

(A2) For the trained $\hat{\phi}$, every pair $(k_1, k_2)$ satisfies $\text{KL}\left(\hat{p}_{\hat{\phi},n}^{(k_2)} \,\|\, \hat{p}_{\hat{\phi},n}^{(k_1)}\right) \le \varepsilon$ a.s. for all $n$.

(A3) All pairs $(k_1, k_2)$ have positive probability in the SC loss.

Under these conditions, we derive the following bound on the martingale residual:

**Theorem 4.2** (Fixed-query martingale induced by SC loss). *Fix an initial context $\mathbf{x}_Q$ and for bounded $f : \mathcal{A} \to \mathbb{R}$, define $\Delta_{n,k}^f(Q) := \mathbb{E}_\phi[f(A_{n+k}) - f(A_{n+1}) \mid \mathcal{F}_n, Q]$. Then, for all $n, k$, we have*

$$\sup_{\|h\|_\infty \le 1, \, \|f\|_\infty \le 1} \left| \mathbb{E}\left[ h(A_{1:n}) \Delta_{n,k}^f(Q) \right] \right| \le C\sqrt{\varepsilon},$$

*where $C > 0$ is a universal constant.*

In particular, one can interpret the above result as $\{\theta_n(\mathbf{x}_Q)\}_{n \ge 0}$ being approximately a martingale, which is connected to the approach of Battiston & Cappello (2025).

The three assumptions clarify the interplay between model capacity and the optimization objective. (A1) requires enough rollouts to guarantee the empirical stability of the predictors. (A2) serves as a post-training condition that reflects the representational capacity of the model; it assumes that the target LLM family is flexible enough to capture approximately martingale dynamics. Under this expressivity condition, the SC objective serves as a proper guide to find such a solution, bounding the horizon-wise discrepancy by $\varepsilon$. (A3) then ensures that this alignment propagates across the relevant horizon range.

Consequently, the theorem tones down the claim to a conditional guarantee: if the model space is flexible enough, the SC tuning objective can successfully find it, thereby bounding the martingale residual. In practice, these conditions are only required to hold over the finite horizon set, making them empirically justifiable.

The detailed proof of Theorem 4.2 is provided in Section C in the appendix. Crucially, while the theorem is stated for a fixed query to establish the pointwise guarantee, our training objective $\mathcal{L}_{\text{SC}}$ minimizes the expected martingale residual over the data distribution $\mathcal{D}$. Consequently, the result extends to unseen queries: by optimizing the expectation, the model learns a generalized mechanism for self-consistency rather than memorizing specific answer trajectories. Thus,

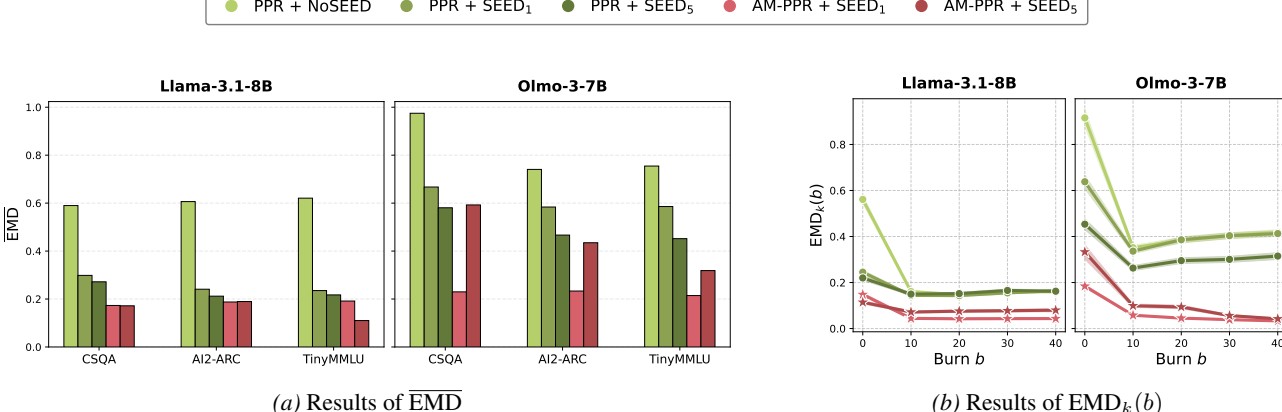

*(a) Results of $\overline{\text{EMD}}$*    *(b) Results of $\text{EMD}_k(b)$*

*Figure 2.* **Analysis of the martingale property violation in PPR.** We evaluated LLAMA-3.1-8B and OLMO-3-7B across 50 questions sampled from three benchmarks (CSQA, AI2-ARC, TinyMMLU) with L1 distance metric. **(a) Expected Mean Drift:** We observe a significant reduction in martingale violation when finetuning with the amortization objective ($\mathcal{L}_{\text{SC}}$). Additionally, providing stochastic seed answers ($m$) consistently improves stability across all models. **(b) Convergence Speed ($\text{EMD}_k(b)$):** We measured the Expected Drift Metric after burning the initial $n$ answers (with a fixed lookahead of $k = 10$) on CSQA benchmark. The shaded region indicates $\pm 1$ standard deviation over the questions. The results demonstrate that both answer seeding and $\mathcal{L}_{\text{SC}}$ training accelerate the stabilization of marginal distributions, significantly reducing the required burn-in period.

for any new query $Q_{\text{test}} \sim \mathcal{D}$, the model is expected to satisfy the $\varepsilon$-approximate martingale property, producing stable zero-shot predictions without the need for inference-time interventions. We refer the reader to Theorem C.2 for the detailed statement and proof of the generalization of Theorem 4.2 to unseen queries.

Finally, while our theory is stated for the SC objective alone, in practice, we occasionally augment optimization with a small score-function (policy-gradient style) correction term to mitigate finite-sample bias in the Monte Carlo estimators $\hat{p}_{\phi,n}^{(k)}$. This correction is used purely as an optimization heuristic and does not alter the theoretical implication. Additional proofs and details can be found in Appendix C.

*Remark.* In practice, we optimize the SC loss using a finite set of horizon pairs, so the implementation does not cover the full $(k_1, k_2)$ range assumed by the theorem. Despite this mismatch, Section 5 shows that the learned model attains approximately martingale behavior in practice since the region of instability is only at the beginning.

# 5. Experiment

We evaluate the ability of Large Language Models to generate coherent martingale sequences within the context of Multiple Choice Question Answering (MCQA). The primary objective of our experiments is to empirically validate that our proposed Stochastic Steering and Amortization successfully mitigate this violation. Furthermore, we assess whether enforcing martingale stability comes at a cost to predictive accuracy. We report performance metrics confirming that our method preserves and often improves the base model's calibration and reliability.

We conducted experiment across two open-source instruct-tuned language models: LLAMA-3.1-8B (Grattafiori et al., 2024), and OLMO-3-7B (Olmo Team et al., 2025). For MCQA benchmark datasets, we used queries from CSQA (Talmor et al., 2019), AI2-ARC (Clark et al., 2018), and TinyMMLU (Polo et al., 2024). Training datasets for the amortization over each language model were constructed from sample queries from the CSQA training split. Further details, including hyperparameters, can be found in Appendix D.

## 5.1. Martingale Property Violation.

**Measuring martingale violations.** Unlike Falck et al. (2024), who quantify violations via an MLE proxy $\bar{\theta}_N$, we measure martingale violations directly in terms of the model distribution $p_\phi$. This is feasible in our MCQA setting, where the full predictive distribution over the discrete answer space is directly accessible, enabling a more faithful assessment of distributional drift.

To this end, we introduce two complementary diagnostics based on *Expected Martingale Drift* (EMD): (i) $\overline{\text{EMD}}$, which summarizes the average martingale violation along the entire answer path, and (ii) $\text{EMD}_k(b)$, which isolates the *early* drift pattern observed in Section 3. Concretely, $\overline{\text{EMD}}$ is the expected distance between the immediate (1-step) prediction $p_{\phi,n}^{(1)}$ and the $k$-step-ahead prediction $p_{\phi,n}^{(k)}$, both conditioned on the current history, averaged over $k$:

$$\overline{\text{EMD}} := \frac{1}{K} \sum_k \mathbb{E}_{Q \sim \mathcal{D}} \left[ \mathbb{E}_{A_{1:n}} \left[ d\left(p_{\phi,n}^{(k)}, p_{\phi,n}^{(1)}\right) \right] \right].$$

We estimate the $k$-step predictions using the same rollout procedure as in the SC loss objective (Definition 4.1), en-

*Table 1.* Accuracy and AUC score on TinyMMLU, AI2-ARC, CSQA inferred by LLAMA-3.1-8B

| Method | TinyMMLU | | AI2-ARC | | CSQA | |
|---|---|---|---|---|---|---|
| | ACC | AUC | ACC | AUC | ACC | AUC |
| PPR + NoSEED | 46.0 | 78.0 | 62.6 | 85.6 | 50.0 | 85.6 |
| PPR + SEED$_1$ | 59.2 | 80.0 | 77.8 | 91.3 | 74.0 | 92.7 |
| PPR + SEED$_5$ | 61.2 | **83.1** | 80.7 | 93.0 | **78.0** | 92.7 |
| AM-PPR + SEED$_1$ | **62.3** | 83.1 | **81.5** | 92.9 | **78.0** | **93.1** |
| AM-PPR + SEED$_5$ | **62.2** | **85.1** | 80.5 | **94.2** | 77.0 | **94.2** |
| Direct-Query | 61.2 | 81.6 | 80.5 | **93.1** | 78.0 | 91.9 |

*Table 2.* ECE, NLL and Brier Score on TinyMMLU, AI2-ARC, CSQA inferred by LLAMA-3.1-8B

| Method | TinyMMLU | | | AI2-ARC | | | CSQA | | |
|---|---|---|---|---|---|---|---|---|---|
| | ECE | NLL | BS | ECE | NLL | BS | ECE | NLL | BS |
| PPR + NoSEED | 0.1763 | 1.2480 | 0.6247 | 0.1623 | 1.1199 | 0.5222 | 0.1273 | 1.5178 | 0.6430 |
| PPR + SEED$_1$ | 0.1537 | **1.0526** | 0.5346 | 0.1695 | **0.8936** | 0.3968 | 0.1522 | 0.9109 | 0.3842 |
| PPR + SEED$_5$ | **0.1278** | 1.0721 | **0.5088** | 0.1259 | 0.7585 | 0.3281 | 0.1967 | **0.8832** | 0.4324 |
| AM-PPR + SEED$_1$ | **0.1229** | 2.2144 | **0.5018** | 0.1943 | 0.9240 | 0.3567 | 0.1139 | 0.9722 | 0.3537 |
| AM-PPR + SEED$_5$ | 0.1568 | 1.2690 | 0.5123 | **0.0845** | 1.2752 | **0.2872** | **0.0674** | **0.7886** | **0.3299** |
| Direct-Query | 0.2457 | 5.4862 | 0.5631 | **0.1251** | 2.4481 | **0.2978** | 0.1072 | 3.0363 | **0.3453** |

suring that evaluation is consistent with the training signal. In practice, instead of averaging over all prefixes $A_{1:n}$, we approximate the expectation by sampling one prefix from the top-$K$ most likely prefixes for each $n \in \mathcal{N}$ (we use $K = 5$ and $\mathcal{N} = \{0, 2, 4, 6, 8\}$). Finally, to focus on prefix-dependent *initial* drift, we define

$$\text{EMD}_k(b) := \mathbb{E}_{Q \sim \mathcal{D}} \big[ \mathbb{E}_{A_{1:n}} \big[ d\big(p_{\phi,n+b}^{(k)}, p_{\phi,n+b}^{(1)}\big) \big] \big],$$

which measures the $k$-step drift at an offset $b$ along the trajectory.

**Results** Figure 2 shows that martingale violations are dramatically reduced after tuning with the SC loss. In particular, Figure 2a indicates that AM-PPR consistently achieves substantially lower $\overline{\text{EMD}}$ value than PPR across all datasets and base models, confirming that our objective effectively suppresses distributional drift during learning. A similar trend appears in Figure 2b: compared to PPR, AM-PPR significantly mitigates the early drift pattern observed in Section 3, providing direct evidence that our method targets the transient martingale violation in the initial regime. Finally, consistent with Section 3, answer seeding reduces martingale violations for PPR; nevertheless, AM-PPR remains strictly better even *after* applying the seeding, showing that SC tuning and answer seeding are complementary and that the best performance is achieved when combining both.

## 5.2. Predictive performance and calibration

Having validated the stability of the belief process, we now evaluate the quality of the resulting martingale posteriors. Since the martingale property guarantees the existence and convergence of a stable limit belief (as discussed in Section 2), we can treat the estimated martingale posterior mean $q^*$ as the model's "true" answer distribution and evaluate it against standard metrics: Accuracy (Acc), AUC, Expected Calibration Error (ECE), Negative Log-Likelihood (NLL), and Brier Score (BS). Refer to Section D.5 to see the detailed definitions of the eval metrics.

**Efficient Estimation via Amortization.** A critical advantage of our method lies in the efficiency of estimation. Because AM-PPR achieves sufficiently low EMD (negligible drift), the initial predictive distribution effectively approximates the infinite-horizon limit. Consequently, we estimate the martingale posterior $q^*$ differently for each method:

- **PPR (Base):** We estimate $q^* \approx \hat{\theta}_N$, requiring a long rollout ($N \gg 1$) to allow the process to stabilize.

- **AM-PPR (Ours):** We estimate $q^* \approx \hat{\theta}_1$, utilizing the immediate next-token distribution.

This highlights the practical utility of our approach: AM-PPR allows us to access the stable belief state zero-shot, eliminating the computational overhead of generating long sequences.

**Results.** We observe that answer seeding is essential for maintaining predictive accuracy comparable to PPR in Table 1. Furthermore, in Table 2, AM-PPR demonstrates improvements in calibration metrics (ECE and NLL) across benchmarks. While standard instruction-tuned models are often overconfident, the martingale-based estimation—particularly when regularized via our amortization objective—yields uncertainty estimates that are better aligned with the empirical correctness of the answers. However, the results of the PPR variants reveal a nuanced relationship: enforcing the martingale property does not monotonically improve calibration. This indicates that the martingale property is merely *necessary* condition for valid Bayesian inference; consequently, minimizing martingale violations guarantees internal consistency, but does not inherently ensure that the model is well-calibrated. Additional experiment results can be found in Appendix E.

## 6. Conclusion

We revisited the question of whether LLM predictive behavior can be meaningfully interpreted through the martingale/Bayesian lens, extending the martingale perspective of Falck et al. (2024) to the most common deployment regime: repeatedly answering a fixed query. By introducing *prompted predictive resampling* and an SC loss that *amortizes* stabilization, we showed that practical LLMs exhibit a transient belief drift that can be substantially reduced. Across standard QA benchmarks, the resulting stabilized predictive distributions improve uncertainty calibration while maintaining accuracy, supporting the view that martingale-style coherence can be operationalized as a useful alignment target for predictive uncertainty.

Our findings also suggest clear limitations and directions for future work. First, reducing martingale violation is not by itself sufficient to guarantee well-calibrated uncertainty: even if the predictive sequence is approximately martingale, this remains only a *necessary* condition for a fully Bayesian interpretation, and additional structure is needed to ensure that calibrated probabilities align with task-ground truth. Second, our method requires prompt engineering to elicit clean sequential resampling behavior, indicating that practical deployment may require careful prompt design. Developing training and inference procedures that produce truly Bayesian-calibrated predictors while minimizing prompt dependence remains an important open problem.

Finally, while our primary evaluation is focused on the MCQA setting to exactly measure drift over a discrete answer simplex, our framework is inherently extensible to richer reasoning trajectories. As a proof of concept, our preliminary experiments demonstrate that martingale consistency successfully emerges when evaluated over the embedding spaces of generated solution-answer pairs. The

results can be found in Appendix F. Ultimately, scaling this framework to analyze and regularize the *stochastic thinking paths* of language models, thereby ensuring representational belief stability across complex multi-step reasoning trajectories, opens up a highly promising avenue for future research.

## Acknowledgements

This work was partly supported by Institute of Information & communications Technology Planning & Evaluation(IITP) grant funded by the Korea government(MSIT) (No.RS-2019-II190075, Artificial Intelligence Graduate School Program(KAIST); No.RS-2022-II220184, Development and Study of AI Technologies to Inexpensively Conform to Evolving Policy on Ethics), and the National Research Foundation of Korea(NRF) grant funded by the Korea government(MSIT) (RS-2022-NR070855). HG was supported by the Institute of Information & Communications Technology Planning & Evaluation(IITP) grant funded by the Korea government(MSIT) (No.RS-2025-02219317, AI Star Fellowship(Kookmin University)) and the National Research Foundation of Korea(NRF) grant funded by the Korea government(MSIT) (RS-2026-25480825).

## Impact Statement

This work significantly advances the reliability of Generative AI by addressing belief drift, a fundamental phenomenon where Large Language Models violate the martingale property and exhibit transient inconsistencies during autoregressive generation. Societally, by enforcing predictive coherence and improving uncertainty calibration without sacrificing accuracy, our approach enhances the trustworthiness of AI systems, a critical prerequisite for their responsible deployment in high-stakes domains such as medical diagnosis, legal analysis, and scientific reasoning.

Environmentally, our proposed *Martingale Amortization* substantially improves inference efficiency; by distilling long-horizon stability into early generation steps via the SC loss, we eliminate the need for computationally expensive repeated sampling or long rollouts to access the model's true belief state. However, it is crucial to note that this method stabilizes the model's internal beliefs regardless of their factual correctness. Consequently, while it mitigates transient inconsistency, it does not inherently correct deep-seated biases or hallucinations, necessitating complementary alignment measures to ensure that the stabilized beliefs remain grounded in truth and safety.

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

## A. Related Work

A growing body of work explains ICL through the lens of Bayesian inference, arguing that a pretrained transformer can *implicitly* infer a latent concept or knowledge variable that summarizes the demonstrations, and then predict new outputs by conditioning on this latent state (Xie et al., 2022; Jiang, 2023; Wang et al., 2024; Ye et al., 2024). Related analyses connect ICL to Bayesian model averaging when demonstrations are treated as exchangeable (Zhang et al., 2023), and to Bayes-optimal prediction in stylized regimes where high-capacity transformers trained with square loss recover Bayes predictors (Akyürek et al., 2022; Panwar et al., 2023). In a particularly clean setting, Xie et al. (2022) proves implicit Bayesian inference for transformers trained on data generated from a Hidden Markov Model; while this likelihood is not i.i.d. exchangeable in general, the analysis suggests that sample-efficient ICL on exchangeable data may require the model to recognize (approximately) exchangeable structure.

Our work is directly motivated by a complementary statistical line: the *predictive modelling paradigm* in Bayesian statistics, which studies whether one-step predictive sequences (without an explicit latent variable) can be represented *as if* they arose from a latent $\theta$ with a prior–likelihood pair (Fong et al., 2023; Fortini & Petrone, 2025). In this framework, exchangeability guarantees the existence of an underlying latent variable, and the stronger condition of c.i.d., equivalently, a martingale property of predictive beliefs, yields a *martingale posterior* that generalizes classical posteriors (Berti et al., 2004; Fong et al., 2023). Leveraging this connection, Falck et al. (2024) derives diagnostics and presents evidence that current LLMs can violate the martingale property on synthetic/toy ICL tasks, questioning the hypothesis that ICL is Bayesian in general. In contrast, we extend the martingale lens beyond synthetic ICL settings to the common task with regime: repeatedly answering a fixed query $Q$. We show that belief drift can be transient yet practically significant, and propose mechanisms to *steer* the induced predictive sequence toward martingale-like behavior: an inference-time answer seeding strategy and a training-time objective (SC loss) that directly targets predictive coherence.

## B. Proofs of Theoretical Statements

This appendix collects proofs for the theoretical claims used throughout the paper. We first show that the martingale property of the predictive belief process is equivalent to the conditional identical distribution (c.i.d.) property of the generated answer sequence, formalizing the intuition that a coherent belief should not depend on how far into the future we look. We then prove a consistency result for our MP estimator: under the martingale property, the empirical MLE fitted to a long resampled trajectory converges almost surely to the limiting belief $\theta_\infty(\mathbf{x}_Q)$.

The first proposition is a structural equivalence. It connects the martingale condition on belief states (Definition 2.1) to a predictive invariance principle: if $\{\theta_n\}$ is a martingale, then conditioning on the current history $A_{1:n}$, the distribution of any future answer $A_{n+k}$ is the same as that of the next answer $A_{n+1}$. In other words, the model's future marginals are time-homogeneous given the past, which is precisely the c.i.d. property for the generated sequence.

**Proposition B.1.** *Let the belief state be $\theta_n(\cdot) := p_\phi(A_{n+1} = \cdot \mid A_{1:n}, \mathbf{x})$ where $A_{1:n} \sim p_\phi(\cdot \mid \mathbf{x})$ for a fixed context $\mathbf{x}$. The following arguments are equivalent.*

*(i)* $\mathbb{E}[\theta_{n+k} \mid A_{1:n}] = \theta_n,$

*(ii)* $A_{1:\infty} \sim p_\phi(\cdot \mid \mathbf{x})$ *is conditionally identically distributed,*

*(iii)* $p_\phi(A_{n+k} = a \mid A_{1:n}, \mathbf{x}) = p_\phi(A_{n+1} = a \mid A_{1:n}, \mathbf{x}) \quad \forall n, k \in \mathbb{N} \quad \forall a \in \mathcal{A}.$

*Proof.* The above results are standard and can be found in Berti et al. (2004).

$\square$

The second proposition justifies the estimation step used in Section 4. Under the categorical likelihood, the MP estimator reduces to the empirical frequency vector along a predictive-resampling trajectory. When the predictive belief process is a martingale, it admits an almost-sure limit $\theta_\infty(\mathbf{x}_Q)$; the result shows that, as the horizon $N$ grows, the empirical frequency estimator converges almost surely to this same limit. This provides the theoretical basis for using long predictive-resampling rollouts (and Monte Carlo averaging across trajectories) to approximate the martingale posterior distribution over $\theta_\infty(\mathbf{x}_Q)$.

**Proposition B.2** (Consistency of the MP estimator). *Fix $\mathbf{x}_Q$ and let $A_{1:\infty} \sim p_\phi(\cdot \mid \mathbf{x}_Q)$ with $\mathcal{F}_n := \sigma(A_{1:n})$. Assume the predictive belief process*

$$\theta_n(\mathbf{x}_Q) := p_\phi(A_{n+1} \in \cdot \mid \mathcal{F}_n, \mathbf{x}_Q)$$

*is a martingale. Let $\bar{\theta}_N^{MP} := \arg\max_{\vartheta \in \mathcal{P}(\mathcal{A})} \log p(A_{1:N} \mid \vartheta, \mathbf{x}_Q)$, where $p(A_{1:N} \mid \vartheta, \mathbf{x}_Q) = \prod_{n=1}^N \vartheta(A_n)$. Then $\bar{\theta}_N^{MP} \to \theta_\infty(\mathbf{x}_Q)$ a.s.*

*Proof.* Let $\mathcal{F}_n := \sigma(A_{1:n})$ and recall that $\mathcal{A}$ is finite.

**Step 1: Closed-form of the MP estimator under the categorical likelihood.** Under the categorical likelihood,

$$p(A_{1:N} \mid \vartheta, \mathbf{x}_Q) = \prod_{n=1}^N \vartheta(A_n), \qquad \vartheta \in \mathcal{P}(\mathcal{A}), \tag{9}$$

the log-likelihood is

$$\log p(A_{1:N} \mid \vartheta, \mathbf{x}_Q) = \sum_{n=1}^N \log \vartheta(A_n) = \sum_{a \in \mathcal{A}} N_a \log \vartheta(a), \tag{10}$$

where $N_a := \sum_{n=1}^N \mathbf{1}\{A_n = a\}$. Maximizing $\sum_a N_a \log \vartheta(a)$ subject to $\sum_a \vartheta(a) = 1$ yields the standard multinomial MLE,

$$\bar{\theta}_N^{MP}(a) = \frac{N_a}{N} = \frac{1}{N} \sum_{n=1}^N \mathbf{1}\{A_n = a\}, \qquad \forall a \in \mathcal{A}. \tag{11}$$

**Step 2: Almost-sure convergence to $\theta_\infty$.** Fix any $a \in \mathcal{A}$ and define $X_n := \mathbf{1}\{A_n = a\}$. By the definition of the predictive belief,

$$\mathbb{E}[X_{n+1} \mid \mathcal{F}_n, \mathbf{x}_Q] = \mathbb{P}_\phi(A_{n+1} = a \mid \mathcal{F}_n, \mathbf{x}_Q) = \theta_n(\mathbf{x}_Q)(a). \tag{12}$$

Let $m_n := \theta_n(\mathbf{x}_Q)(a)$ and define the martingale differences

$$D_{n+1} := X_{n+1} - m_n. \tag{13}$$

Then $\mathbb{E}[D_{n+1} \mid \mathcal{F}_n, \mathbf{x}_Q] = 0$ and $|D_{n+1}| \leq 1$. Let $S_N := \sum_{n=0}^{N-1} D_{n+1}$. Then $(S_N)$ is a martingale, and its increments satisfy $|S_{N+1} - S_N| = |D_{N+1}| \leq 1$ a.s..

By Azuma–Hoeffding (Azuma, 1967), for any $\varepsilon > 0$,

$$\mathbb{P}\left( \left| \frac{S_N}{N} \right| \geq \varepsilon \right) \leq 2 \exp\left( -\frac{N\varepsilon^2}{2} \right). \tag{14}$$

Since $\sum_{N=1}^\infty 2\exp(-\frac{N\varepsilon^2}{2}) < \infty$, the Borel–Cantelli lemma (Williams, 1991) implies $\frac{S_N}{N} \to 0$ almost surely.

Now decompose the empirical mean:

$$\bar{\theta}_N^{MP}(a) = \frac{1}{N} \sum_{n=1}^N X_n = \frac{1}{N} \sum_{n=1}^N m_{n-1} + \frac{1}{N} \sum_{n=1}^N (X_n - m_{n-1}) = \frac{1}{N} \sum_{n=0}^{N-1} m_n + \frac{S_N}{N}.$$

By assumption and Doob's convergence (Doob, 1949), $(\theta_n(\mathbf{x}_Q))_{n \geq 0}$ is a martingale and $\theta_n(\mathbf{x}_Q) \to \theta_\infty(\mathbf{x}_Q)$ almost surely; hence $m_n \to \theta_\infty(\mathbf{x}_Q)(a)$ almost surely. By Cesàro convergence (Williams, 1991),

$$\frac{1}{N} \sum_{n=0}^{N-1} m_n \to \theta_\infty(\mathbf{x}_Q)(a) \quad \text{a.s.}$$

Combining with $\frac{S_N}{N} \to 0$ a.s., we obtain

$$\bar{\theta}_N^{MP}(a) \to \theta_\infty(\mathbf{x}_Q)(a) \quad \text{a.s.}$$

Since $\mathcal{A}$ is finite, the above convergence holds simultaneously for all $a \in \mathcal{A}$ (on the intersection of finitely many probability-one events), which implies $\bar{\theta}_N^{MP} \to \theta_\infty(\mathbf{x}_Q)$ a.s. as vectors in $\mathcal{P}(\mathcal{A})$. $\qquad\square$

## C. Theoretical Justification of Self-Consistency (SC) Loss

Here, we provide a theoretical justification of the proposed SC loss by establishing its connection to martingale belief dynamics. We first analyze the fixed-query setting and show that, if the SC loss is sufficiently small, then the predictive belief process induced by the language model satisfies an approximate martingale property. This result formalizes the intuition that matching short-horizon predictions to long-horizon predictions enforces internal self-consistency and stabilizes the model's belief dynamics without requiring inference-time interventions.

**Theorem C.1** (Fixed-query martingale residual bound induced by SC loss). *Fix an initial context $\mathbf{x}_Q$ and let $(A_n)_{n \geq 1}$ be the sequence of answers generated by a language model $p_\phi$ when repeatedly prompted with $\mathbf{x}_Q$. Let $\mathcal{F}_n := \sigma(A_{1:n})$ denote the filtration generated by the answer history. For $k \geq 1$, define*

$$p_{\phi,n}^{(k)}(\cdot) := \mathbb{P}_\phi(A_{n+k} \in \cdot \mid \mathcal{F}_n, \mathbf{x}_Q). \tag{15}$$

*For any bounded test function $f : \mathcal{A} \to \mathbb{R}$, define*

$$\Delta_{n,k}^f(\mathbf{x}_Q) := \mathbb{E}_\phi\big[f(A_{n+k}) - f(A_{n+1}) \mid \mathcal{F}_n, \mathbf{x}_Q\big]. \tag{16}$$

*Assume that:*

*(A1) $\hat{p}_{\phi,n}^{(k)}$ is uniformly consistent at $p_{\phi,n}^{(k)}$ over $n, k$ as the number of rollouts grows.*

*(A2) For the trained parameter $\hat{\phi}$, every horizon pair $(k_1, k_2)$ satisfies*

$$\mathrm{KL}\Big(\hat{p}_{\hat{\phi},n}^{(k_2)} \,\|\, \hat{p}_{\hat{\phi},n}^{(k_1)}\Big) \leq \varepsilon \quad \text{a.s. for all } n. \tag{17}$$

*(A3) Every horizon pair $(k_1, k_2)$ appears in the SC loss with positive probability.*

*Then, for all $n, k$, we have*

$$\sup_{\|h\|_\infty \leq 1,\, \|f\|_\infty \leq 1} \Big|\mathbb{E}\big[h(A_{1:n})\, \Delta_{n,k}^f(\mathbf{x}_Q)\big]\Big| \leq C\sqrt{\varepsilon}, \tag{18}$$

*where $C > 0$ is a universal constant.*

*Proof.* The SC loss compares horizon-wise predictive distributions via cross-entropy with a stop-gradient target. Using the identity

$$H(p, q) = H(p) + \mathrm{KL}(p\|q), \tag{19}$$

each term of the SC loss reduces, up to an additive constant independent of $\phi$, to a KL divergence $\mathrm{KL}(\hat{p}_{\phi,n}^{(k_2)}\|\hat{p}_{\phi,n}^{(k_1)})$.

By Assumption (A2), for the trained parameter $\hat{\phi}$ we have, for all $n$ and all horizon pairs $(k_1, k_2)$,

$$\mathrm{KL}\Big(\hat{p}_{\hat{\phi},n}^{(k_2)} \,\|\, \hat{p}_{\hat{\phi},n}^{(k_1)}\Big) \leq \varepsilon \quad \text{a.s.} \tag{20}$$

By Assumption (A1), uniform consistency implies that the corresponding bound carries over to the true predictive distributions:

$$\mathrm{KL}\Big(p_{\hat{\phi},n}^{(k_2)} \,\|\, p_{\hat{\phi},n}^{(k_1)}\Big) \leq \varepsilon \quad \text{a.s.} \tag{21}$$

Fix $n$ and $k$ such that the horizon pair $(n + 1, n + k)$ is of interest and appears in the SC loss with positive probability (Assumption (A3)). Applying Pinsker's inequality (Pinsker, 1964) yields

$$\big\|p_{\hat{\phi},n}^{(k)} - p_{\hat{\phi},n}^{(1)}\big\|_{\mathrm{TV}} \leq \sqrt{\tfrac{1}{2}\mathrm{KL}\Big(p_{\hat{\phi},n}^{(k)} \,\|\, p_{\hat{\phi},n}^{(1)}\Big)} \leq \sqrt{\tfrac{\varepsilon}{2}} \quad \text{a.s.} \tag{22}$$

Conditioning on $\mathcal{F}_n$ and $Q$, the variational characterization of total variation gives, for any $\|f\|_\infty \leq 1$,

$$|\Delta_{n,k}^f(\mathbf{x}_Q)| \leq 2\big\|p_{\hat{\phi},n}^{(k)}(\mathbf{x}_Q) - p_{\hat{\phi},n}^{(1)}(\mathbf{x}_Q)\big\|_{\mathrm{TV}} \leq \sqrt{2}\,\sqrt{\varepsilon} \quad \text{a.s.} \tag{23}$$

Finally, for any bounded $\mathcal{F}_n$-measurable $h$ with $\|h\|_\infty \leq 1$,

$$\left| \mathbb{E}\big[ h(A_{1:n}) \, \Delta_{n,k}^f(\mathbf{x}_Q) \big] \right| \leq \mathbb{E}\big[ |\Delta_{n,k}^f(\mathbf{x}_Q)| \big] \leq \sqrt{2}\,\sqrt{\varepsilon}. \tag{24}$$

Taking the supremum over $h$ and $f$ completes the proof. $\qquad\square$

Now, we extend the analysis to the generalization setting and study how the martingale property induced by the SC loss transfers to unseen queries. We show that, under standard generalization assumptions, minimizing the empirical SC loss over a finite set of training queries yields approximate martingale behavior in expectation for new queries drawn from the same distribution. This result ensures that SC loss does not merely enforce self-consistency on memorized prompts but instead induces a transferable mechanism for stabilizing belief dynamics.

**Theorem C.2** (Generalization of martingale property by SC loss). *Let $\mathcal{D}$ be a distribution over contexts $\mathbf{x}_Q$. Let $\mathbf{x}_Q^{(1)}, \dots, \mathbf{x}_Q^{(m)} \sim \mathcal{D}$ be i.i.d. training queries, and let $\hat{\phi}$ be a parameter obtained by minimizing the empirical SC loss. Let $\widehat{\mathcal{L}}_{\mathrm{SC}}(\phi)$ denote the empirical SC loss over the training queries $\mathbf{x}_Q^{(1)}, \dots, \mathbf{x}_Q^{(m)}$, i.e.,*

$$\widehat{\mathcal{L}}_{\mathrm{SC}}(\phi) := \frac{1}{m} \sum_{i=1}^{m} \mathcal{L}_{\mathrm{SC}}(\phi; \mathbf{x}_Q^{(i)}), \tag{25}$$

*where $\mathcal{L}_{\mathrm{SC}}(\phi; \mathbf{x}_Q)$ is the per-query SC loss in Definition 4.1 (with the expectation over horizon pairs and rollouts as specified there). For any query $\mathbf{x}_Q$, define $\mathcal{F}_n := \sigma(A_{1:n})$ and*

$$\theta_n(\mathbf{x}_Q) := p_{\hat{\phi}}(\cdot \mid \mathcal{F}_n, \mathbf{x}_Q) \in \Delta(\mathcal{A}). \tag{26}$$

*For any bounded $f : \mathcal{A} \to \mathbb{R}$, define*

$$\Delta_{n,k} f(\mathbf{x}_Q) := \mathbb{E}\big[ f(A_{n+k}) \mid \mathcal{F}_n, \mathbf{x}_Q \big] - \mathbb{E}\big[ f(A_{n+1}) \mid \mathcal{F}_n, \mathbf{x}_Q \big]. \tag{27}$$

*Assume that:*

*(A1) The empirical predictors $\hat{p}_{\phi,n}^{(k)}(\mathbf{x}_Q)$ are uniformly consistent at $p_{\phi,n}^{(k)}(\mathbf{x}_Q)$ over $n, k, \mathbf{x}_Q$ as the number of rollouts grows.*

*(A2) There exists a function $\mathrm{Gen}(m, \delta)$ with $\mathrm{Gen}(m, \delta) \to 0$ as $m \to \infty$ such that, with probability at least $1 - \delta$ over $\mathbf{x}_Q^{(1)}, \dots, \mathbf{x}_Q^{(m)} \sim \mathcal{D}$,*

$$\mathcal{L}_{\mathrm{SC}}(\hat{\phi}) \leq \widehat{\mathcal{L}}_{\mathrm{SC}}(\hat{\phi}) + \mathrm{Gen}(m, \delta). \tag{28}$$

*(A3) Every horizon pair $(k_1, k_2)$ appears in the SC loss with positive probability.*

*Here, $\mathrm{Gen}(m, \delta)$ denotes a standard generalization bound for the SC loss, such as one induced by uniform convergence or Rademacher complexity arguments, which upper-bounds the gap between the empirical and population SC losses and vanishes as the number of training queries $m$ increases. Then for any $n, k$, we have*

$$\mathbb{E}_{Q \sim \mathcal{D}} \left[ \sup_{\substack{\|h\|_\infty \leq 1 \\ \|f\|_\infty \leq 1}} \left| \mathbb{E}\big[ h(A_{1:n}) \, \Delta_{n,k} f(\mathbf{x}_Q) \,\big|\, \mathbf{x}_Q \big] \right| \right] \leq C(k) \sqrt{\widehat{\mathcal{L}}_{\mathrm{SC}}(\hat{\phi}) + \mathrm{Gen}(m, \delta)}, \tag{29}$$

*where $C(k) > 0$ depends on $k$.*

*Proof.* Fix $n$ and a horizon $k$ such that the pair $(n+1, n+k)$ is of interest. Recall that the SC loss (Definition 4.1) compares horizon-wise predictive distributions via cross-entropy with a stop-gradient target. Using the decomposition

$$H(p, q) = H(p) + \mathrm{KL}(p \| q), \tag{30}$$

each term of the SC loss reduces, up to an additive constant independent of $\phi$, to a KL divergence between horizon-wise predictors.

Let $\pi$ denote the sampling distribution over horizon pairs used in the SC loss (Definition 4.1). By Assumption (A3), $\pi(1,k) > 0$. Write the population SC loss as

$$\mathcal{L}_{\mathrm{SC}}(\hat{\phi}) \overset{\mathrm{const}}{=} \mathbb{E}_{(u,v)\sim\pi} \, \mathbb{E}_{\mathbf{x}_Q\sim\mathcal{D}} \Big[ \mathrm{KL}\big( \hat{p}_{\hat{\phi},n}^{(v)}(\mathbf{x}_Q) \, \big\| \, \hat{p}_{\hat{\phi},n}^{(u)}(\mathbf{x}_Q) \big) \Big] \tag{31}$$

where $\overset{\mathrm{const}}{=}$ denote the equal up to constant. Since $\mathrm{KL}(\cdot\|\cdot) \geq 0$, we can lower bound the above expectation by the contribution of the single pair $(u,v) = (1,k)$:

$$\mathcal{L}_{\mathrm{SC}}(\hat{\phi}) \; \geq \; \pi(1,k) \, \mathbb{E}_{\mathbf{x}_Q\sim\mathcal{D}} \Big[ \mathrm{KL}\big( \hat{p}_{\hat{\phi},n}^{(k)}(\mathbf{x}_Q) \, \big\| \, \hat{p}_{\hat{\phi},n}^{(1)}(\mathbf{x}_Q) \big) \Big]. \tag{32}$$

Rearranging yields

$$\mathbb{E}_{\mathbf{x}_Q\sim\mathcal{D}} \Big[ \mathrm{KL}\big( \hat{p}_{\hat{\phi},n}^{(k)}(\mathbf{x}_Q) \, \big\| \, \hat{p}_{\hat{\phi},n}^{(1)}(\mathbf{x}_Q) \big) \Big] \; \leq \; \frac{1}{\pi(1,k)} \, \mathcal{L}_{\mathrm{SC}}(\hat{\phi}). \tag{33}$$

By Assumption (A1), the empirical predictors $\hat{p}_{\hat{\phi},n}^{(k)}(\mathbf{x}_Q)$ are uniformly consistent for the corresponding conditional predictive distributions $p_{\hat{\phi},n}^{(k)}(\mathbf{x}_Q)$ as rollouts grow. Thus, passing to the large-rollout limit in Equation (33) yields the analogous bound for the true horizon predictors:

$$\mathbb{E}_{\mathbf{x}_Q\sim\mathcal{D}} \Big[ \mathrm{KL}\big( p_{\hat{\phi},n}^{(k)}(\mathbf{x}_Q) \, \big\| \, p_{\hat{\phi},n}^{(1)}(\mathbf{x}_Q) \big) \Big] \; \leq \; \frac{1}{\pi(1,k)} \, \mathcal{L}_{\mathrm{SC}}(\hat{\phi}). \tag{34}$$

Next, apply Pinsker's inequality (Pinsker, 1964) and Jensen's inequality:

$$
\begin{aligned}
\mathbb{E}_{\mathbf{x}_Q\sim\mathcal{D}} \Big[ \big\| p_{\hat{\phi},n}^{(k)}(\mathbf{x}_Q) - p_{\hat{\phi},n}^{(1)}(\mathbf{x}_Q) \big\|_{\mathrm{TV}} \Big] &\leq \mathbb{E}_{\mathbf{x}_Q\sim\mathcal{D}} \Big[ \sqrt{\tfrac{1}{2} \, \mathrm{KL}\big( p_{\hat{\phi},n}^{(k)}(\mathbf{x}_Q) \, \big\| \, p_{\hat{\phi},n}^{(1)}(\mathbf{x}_Q) \big)} \Big] \\
&\leq \sqrt{\tfrac{1}{2} \, \mathbb{E}_{\mathbf{x}_Q\sim\mathcal{D}} \Big[ \mathrm{KL}\big( p_{\hat{\phi},n}^{(k)}(\mathbf{x}_Q) \, \big\| \, p_{\hat{\phi},n}^{(1)}(\mathbf{x}_Q) \big) \Big]} \\
&\leq \sqrt{\frac{1}{2\pi(1,k)} \, \mathcal{L}_{\mathrm{SC}}(\hat{\phi})},
\end{aligned}
\tag{35}
$$

where the last inequality uses Equation (34).

Now fix $Q$ and condition on $\mathcal{F}_n$. By the variational characterization of total variation (Le Cam & Yang, 2000), for any bounded $f$ with $\|f\|_\infty \leq 1$,

$$\big| \Delta_{n,k} f(\mathbf{x}_Q) \big| = \Big| \mathbb{E}[f(A_{n+k}) \mid \mathcal{F}_n, \mathbf{x}_Q] - \mathbb{E}[f(A_{n+1}) \mid \mathcal{F}_n, \mathbf{x}_Q] \Big| \leq 2 \big\| p_{\hat{\phi},n}^{(k)}(\mathbf{x}_Q) - p_{\hat{\phi},n}^{(1)}(\mathbf{x}_Q) \big\|_{\mathrm{TV}}. \tag{36}$$

Let $h = h(A_{1:n})$ be any bounded $\mathcal{F}_n$-measurable function with $\|h\|_\infty \leq 1$. Then

$$\Big| \mathbb{E}\big[ h(A_{1:n}) \, \Delta_{n,k} f(\mathbf{x}_Q) \, \big| \, \mathbf{x}_Q \big] \Big| \leq \mathbb{E}\big[ |\Delta_{n,k} f(\mathbf{x}_Q)| \, \big| \, \mathbf{x}_Q \big]. \tag{37}$$

Taking $\sup_{\|h\|_\infty \leq 1}$ and $\sup_{\|f\|_\infty \leq 1}$ and then expectation over $Q \sim \mathcal{D}$, we obtain

$$
\begin{aligned}
\mathbb{E}_{\mathbf{x}_Q\sim\mathcal{D}} \Big[ \sup_{\substack{\|h\|_\infty \leq 1 \\ \|f\|_\infty \leq 1}} \Big| \mathbb{E}\big[ h(A_{1:n}) \, \Delta_{n,k} f(\mathbf{x}_Q) \, \big| \, \mathbf{x}_Q \big] \Big| \Big] &\leq 2 \, \mathbb{E}_{\mathbf{x}_Q\sim\mathcal{D}} \Big[ \big\| p_{\hat{\phi},n}^{(k)}(\mathbf{x}_Q) - p_{\hat{\phi},n}^{(1)}(\mathbf{x}_Q) \big\|_{\mathrm{TV}} \Big] \\
&\leq \sqrt{\frac{2}{\pi(1,k)} \, \mathcal{L}_{\mathrm{SC}}(\hat{\phi})},
\end{aligned}
\tag{38}
$$

where the last inequality uses Equation (35).

Finally, apply the generalization bound in Assumption (A2): with probability at least $1 - \delta$,

$$\mathcal{L}_{\mathrm{SC}}(\hat{\phi}) \; \leq \; \widehat{\mathcal{L}}_{\mathrm{SC}}(\hat{\phi}) + \mathrm{Gen}(m, \delta). \tag{39}$$

Substituting this into Equation (38) yields the desired result with

$$C(k) = \sqrt{\frac{2}{\pi(1,k)}}. \tag{40}$$

$\square$

Now we discuss the bias-correction term in the SC loss. As described in Appendix 4.2, we include this term to remove finite-sample bias and ensure that the objective is an unbiased estimator of the target quantity. Corollary C.3 formalizes this guarantee by showing that, with the proposed correction, the SC loss becomes unbiased.

**Corollary C.3.** *(Bias Correction of SC Loss) Let $\mathbf{x}_Q$ be a fixed query per batch, where batch inputs are denoted as $\mathcal{B} := \{a_{n+1:n+m}^{(j)}\} \sim p_\phi(\cdot \mid a_{1:n}, \mathbf{x}_Q)$. Then our loss function $\mathcal{L}_{SC}(\phi)$ has a biased gradient estimator, and thus, we correct the bias via additional term:*

$$\mathcal{L}(\phi; \mathcal{B}) := \mathcal{L}_{SC}(\phi; \mathcal{B}) + \beta \mathcal{L}_{pg}(\phi; \mathcal{B})$$

*where $\mathcal{L}(\phi)$ has an unbiased gradient estimator, and the correction loss is defined as:*

$$\mathcal{L}_{pg}(\phi; \mathcal{B}) := clip(A, -c_{adv}, -c_{adv})\overline{\log p_\phi}(a_{n+1:n+m} \mid a_{1:n}, \mathbf{x}_Q)$$

*Proof.* The derivation of the bias correction term follows the same logic as policy gradient bias correction (Sutton et al., 1999). $\square$

# D. Additional Experiment Details

All experiments were implemented using the Hugging Face transformers library (Wolf et al., 2020). Inference and training were both conducted on a workstation equipped with two NVIDIA RTX A6000 GPUs (48GB VRAM each). To ensure high numerical precision while optimizing memory throughput, all model weights and activations were loaded in Bfloat16 (BF16) precision.

## D.1. Datasets and Models

**Datasets**    We use CSQA (Talmor et al., 2019), TinyMMLU (Polo et al., 2024), AI2-ARC (Clark et al., 2018) for evaluation. We specifically select tasks with more than three multiple-choice options where baselines achieve at least 50% accuracy; this ensures we avoid degenerate regimes where sampled answer sequences are dominated by near-uniform random noise.

- CSQA [1]: A 5-way multiple-choice commonsense reasoning dataset which consists of $9,741, 1,221$, and $1,141$ number of training, validation, and test data, respectively. We randomly sampled 200 questions from the validation split to ensure access to ground-truth labels for the evaluation.

- AI2-ARC-Challenge [2]: We utilize the "Challenge" subset of AI2-ARC, which contains questions with 4-way multiple choices that simple retrieval and word co-occurrence algorithms fail to solve. This dataset consists of $1,119, 299$, and $1,172$ number of training, validation, and test data, respectively. We randomly sampled 200 questions from this set for the evaluation.

- TinyMMLU [3]: Part of the TinyBenchmarks suite (Polo et al., 2024), this dataset enables a reliable LLM evaluation using a compressed subset of MMLU, consisting of 100 questions with 4-way multiple choices. We utilize the full TinyMMLU set for the evaluation.

During preprocessing, a small number of samples were excluded if the model failed to follow basic formatting instructions. We applied a strict quality-control filter during preprocessing: any question where more than 50% of the sampled generation paths failed to adhere to the required response format was excluded from our analysis. This ensures that our evaluation of model reasoning is not confounded by catastrophic instruction-following failures.

**Models**    We evaluate our methods using the instruction-tuned variants of two widely-adopted open-source Large Language Models: LLAMA-3-8B-INSTRUCT (Grattafiori et al., 2024) and OLMO-3-7B-INSTRUCT (Olmo Team et al., 2025). As discussed in Section 4, the use of instruction-tuned weights is critical for ensuring the models adhere to the structured response formats required for automated evaluation.

- LLAMA-3.1-8B-INSTRUCT: https://huggingface.co/meta-llama/Llama-3.1-8B-Instruct, licensed under Llama 3.1 Community License. [4]

- OLMO-3-7B-INSTRUCT: https://huggingface.co/allenai/Olmo-3-7B-Instruct, licensed under Apache license 2.0. [5]

## D.2. Martingale Amortization

We fine-tune the pretrained models with Fully Sharded Data Parallel (FSDP) and Low-Rank Adaptation (Hu et al., 2022) on two NVIDIA RTX A6000 GPUs, total 32 effective trajectories of sequences. Each training example corresponds to a single rollout trajectory for a fixed query, and the training set is organized into *groups* by initial context $\mathbf{x}_Q$. For each query($Q$) in the CSQA training split and possible subsequence of rollouts($A_{1:n}$), we generate multiple independent rollout sequences from the pretrained model and store them as separate rows sharing the same group identifier. During training, each optimizer update uses only trajectories from a single group (i.e., a single query) so that the Monte Carlo estimators

---

[1]https://huggingface.co/datasets/tau/commonsense_qa
[2]https://huggingface.co/datasets/allenai/ai2_arc
[3]https://huggingface.co/datasets/tinyBenchmarks/tinyMMLU
[4]https://www.llama.com/llama3_1/license/
[5]https://choosealicense.com/licenses/apache-2.0/

$\hat{p}_{\phi,n}^{(k)}$ can be formed by averaging over multiple rollouts of the same conditional histories. Total 32 effective trajectories were used per optimizer update. The following table summarizes the key hyperparameters used in LLAMA-3.1-8B-INSTRUCT and OLMO-3-7B-INSTRUCT.

### D.3. Hyperparameters

**Hyperparameters.** We summarize the hyperparameters used for training with our objective in two tables. Table 3 reports the settings shared across both backbone models, LLAMA-3.1-8B and OLMO-3-7B, including the common optimization and training configurations. In contrast, Table 4 lists model-specific hyperparameters that are not shared between the two backbones (e.g., settings adjusted to accommodate differences in architecture, tokenizer, or training stability). Together, these tables fully specify the hyperparameter choices used in our experiments.

*Table 3.* Key hyperparameter values for training

| Category | Hyperparameter | Value |
|---|---|---|
| LoRA | Rank ($r$) | 16 |
| | Alpha ($\alpha$) | 8 |
| | Dropout | 0.05 |
| | Target Modules | All Linear Layers |
| Optimization | Optimizer | AdamW |
| | LR Scheduler | Cosine |
| | Weight Decay | 0.01 |
| | Warmup Steps | 5 |
| Training | Micro Batch Size (per device) | 4 |
| | Gradient Accumulation Steps | 4 |
| | Mixed Precision | BF16 |
| | Epoch | 2 |

*Table 4.* Martingale-amortization training settings for each models

| Category | Hyperparameter | LLAMA-3.1-8B-INSTRUCT | OLMO-3-7B-INSTRUCT |
|---|---|---|---|
| Training | Learning rate | $2 \times 10^{-6}$ | $1 \times 10^{-4}$ |
| | # query groups | 20 | 12 |
| | Prefix lengths ($\mathcal{N}$) | $\{0, 2, 4, 6, 8\}$ | $\{0, 2\}$ |
| | Optimizer steps per group $U$ | 1 | 4 |
| $\mathcal{L}_{\text{SC}}$ Computation | PG correction weight($\beta$) | 0.01 | 0.01 |
| | Advantage clamp | 5.0 | 5.0 |

### D.4. Prompts

In this section, we summarize the prompts used to generate answer sequences for training and evaluation. Across all experiments, we primarily use two prompt types: (i) a *predictive-resampling* prompt that, given a question $Q$ and a fixed seed answer history $a_{1:n}$, sequentially samples the continuation $A_{n+1:n+k}$; and (ii) a *seeding* prompt that, given only $Q$, samples an initial seed history $a_{1:n}$.

Figure 3 and Figure 4 show the predictive-resampling prompts used for LLAMA-3.1-8B and OLMO-3-7B, respectively, which condition on $(Q, a_{1:n})$ and generate $A_{n+1:n+k}$ in a sequential manner. In contrast, Figure 5 presents the seeding prompt used to sample $a_{1:n}$, which is shared across all models and datasets for consistency.

---

**(LLAMA-3.1-8B) PPR Prompt**

```
## Task
You are an expert in multiple-choice QA.
Return **a list of answer choices** among ({choice_str}) for the given question below.

### Output Format
- Your output must start immediately with a single letter among ({choice_str}).
- Your seperator is a single newline(\n).
- \n must be appear **only once** after the each choice.
- Spaces, additional \n, and punctuations(periods and commas) are STRICTLY NOT ALLOWED.
- You must output total 100 letters.
- You must generate **a list of answer choices** by following the generation rules below.

### Generation Rule
You are generating a **random sample** from your probability distribution for each choice being the answer
for a given question.
Follow the steps.
(STEP 1) First, assign probability weight on each choice being an answer.
        - If a choice is likely to be an answer, it must have higher probability.
        - Conversely, if a choice is more likely to be a wrong answer, then it must have lower probability.
        - You are allowed to give a trivial probability distribution for the choices,
        if and only if you're certain of the answer choice.
        (i.e. multinomial distribution on {num_choices}-dim answer choices.)

(STEP 2) Write down each line.
        Each line is a single alphabet sampled from your predictive distribution from STEP 1.
        (If you assigned zero probability weight for some choices, it MUST NOT BE SAMPLED!)
        - Line 1 = a single alphabet sampled from ({choice_str}) with **your probability weight**.
        - Line $i$ ($2 <= i <= 100$) = a single alphabet independently sampled from ({choice_str}) with
        **your probability weight**.
        - You must not condition on your previous (Line 1 ~ $i-1$) answers.
        Your answer must be an i.i.d. sample from your distribution.

#### Generation examples
* If you think D is an answer for given question with 100% probability, then output might be :
D\nD\nD\nD\nD\nD\n ...
* If you think B is the most plausible answer, but E can might also be an answer with small probability,
then output might be: B\nB\nB\nB\nE\nB\nB ...
* If you think either both A or C can be an answer with high probability, and the others(B, D, E, etc.) cannot
be the answer, then output might be : C\nA\nC\nC\nA\nA\n ... or A\nC\nC\nA\nA\nC\n ...

Now, generate the 100-line answer list for the given question below.
You must follow the output format and the generating rules!

Question:
```
```

*Figure 3.* (**LLAMA-3.1-8B**) Predictive-Resampling Prompts used for LLAMA-3.1-8B. Here, we give some generation examples to guide the model to generate according to the predictive distribution while following the answer format.

---

**(OLMO-3-7B, OLMO-3.1-32B) PPR Prompt**

```
You are a Synthetic Data Generation Expert.

## Task
Below is a multiple-choice question.
Generate {gen_len} independent responses to the question.
The responses must reflect your beliefs about the most likely answer distribution.

### Simulation Logic

    1. Analyze: Evaluate the question and its answer choices.
    2. Distribute: Assign a probability weight to each option.
        - If you believe an option is more likely to be correct, assign it a higher probability.
        - Trivial answer choice is only allowed when you are absolutely certain of the answer.
            **Conversely, if you are uncertain, you must assign non-trivial probabilities to multiple choices
            for the candidates!!**
    3. Generate: Create {gen_len} answers. For each line, decide the answer independently.
        - Samples must be generated independently; you must not condition on your previous answers.
        - You must NOT create repeating or cyclic patterns
            (e.g., ABAB, ABBABB, AABB, etc.).
            If such a pattern emerges, restart the generation internally.

Use the following format to respond:
- Output ONLY the list of answers separated by newlines (e.g., A\nB\nA...) without any explanations.
- Do not include the count, the probabilities, or any introductory text.

Question:
```
```

*Figure 4.* **(OLMO-3-7B, OLMO-3.1-32B)** Predictive-Resampling Prompts used for OLMO-3-7B. Here, we design a prompt for OLMO-3-7B that places stronger emphasis on enforcing the exact output format, addressing the model's tendency to occasionally deviate from the required answer formatting.

---

**Direct-Query Prompt**

```
'''
Below is a multiple choice question.
Return only one letter among {", ".join(list(string.ascii_uppercase[:num_choices]))} which corresponds
to your answer, without any spaces or newlines.
Follow the below format:
"Answer: [Write your answer choice letter here]"
```
```

*Figure 5.* **Direct-Query Prompt.** This prompt is used to sample the seed answers. We keep it intentionally simple so that it generalizes reliably across all models and datasets.

## D.5. Evaluation Metrics

In this section, we define the evaluation metrics used throughout the paper. Let a probabilistic classifier output a categorical distribution $\mathbf{p}(\mathbf{x}) \in [0,1]^K$ over $K$ classes, with $\sum_{k=1}^{K} p^{(k)}(\mathbf{x}) = 1$. Given a labeled dataset $\mathcal{D} = \{(\mathbf{x}_i, y_i)\}_{i=1}^{N}$ with $y_i \in \{1, \ldots, K\}$, define the predicted label and confidence as $\hat{y}(\mathbf{x}) = \arg\max_k p^{(k)}(\mathbf{x})$ and $c(\mathbf{x}) = \max_k p^{(k)}(\mathbf{x})$. We report the following standard metrics:

- **Accuracy (ACC).**

$$\mathrm{ACC}(\mathcal{D}) := \frac{1}{N} \sum_{i=1}^{N} \mathbb{1}\{\hat{y}(\mathbf{x}_i) = y_i\}. \tag{41}$$

- **Negative log-likelihood (NLL).**

$$\mathrm{NLL}(\mathcal{D}) := \frac{1}{N} \sum_{i=1}^{N} \left(-\log p^{(y_i)}(\mathbf{x}_i)\right). \tag{42}$$

- **Brier score (BS; Brier, 1950).** Let $\mathbf{y}_i \in \{0,1\}^K$ be the one-hot encoding of $y_i$.

$$\mathrm{BS}(\mathcal{D}) := \frac{1}{N} \sum_{i=1}^{N} \|\mathbf{p}(\mathbf{x}_i) - \mathbf{y}_i\|_2^2. \tag{43}$$

- **Expected calibration error (ECE; Naeini et al., 2015).** Partition the confidence interval $[0,1]$ into $N_{\mathrm{bin}}$ bins $I_b = \big((b-1)/N_{\mathrm{bin}},\, b/N_{\mathrm{bin}}\big]$ for $b = 1, \ldots, N_{\mathrm{bin}}$, and let $B_b = \{i : c(\mathbf{x}_i) \in I_b\}$ with $n_b = |B_b|$. Define the bin accuracy and mean confidence as

$$\mathrm{acc}(B_b) := \frac{1}{n_b} \sum_{i \in B_b} \mathbb{1}\{\hat{y}(\mathbf{x}_i) = y_i\}, \qquad \mathrm{conf}(B_b) := \frac{1}{n_b} \sum_{i \in B_b} c(\mathbf{x}_i), \tag{44}$$

(with the convention that empty bins contribute 0). Then

$$\mathrm{ECE}(\mathcal{D}, N_{\mathrm{bin}}) := \sum_{b=1}^{N_{\mathrm{bin}}} \frac{n_b}{N} \left|\mathrm{acc}(B_b) - \mathrm{conf}(B_b)\right|. \tag{45}$$

We use $N_{\mathrm{bin}} = 15$ throughout.

- **Area under the ROC curve (AUROC).** We compute AUROC for *correctness detection* using confidence as a score: assign a binary label $t_i = \mathbb{1}\{\hat{y}(\mathbf{x}_i) = y_i\}$ and a score $s_i = c(\mathbf{x}_i)$. AUROC is the probability that a randomly chosen correct example is assigned a higher score than a randomly chosen incorrect example:

$$\mathrm{AUROC} := \mathbb{P}\big(s^+ > s^-\big) + \tfrac{1}{2} \mathbb{P}\big(s^+ = s^-\big), \tag{46}$$

where $(s^+, s^-)$ are scores drawn from the sets $\{s_i : t_i = 1\}$ and $\{s_i : t_i = 0\}$, respectively.

# E. Additional Experimental Results

In this section, we provide a comprehensive breakdown of the martingale violation analysis discussed in Appendix 5. We extend our evaluation of PPR and AM-PPR across the language models on all three MCQA benchmarks (CSQA, AI2-ARC, and TinyMMLU). To rigorously quantify the deviation from the martingale property (i.e., the drift between the immediate predictive distribution and the stabilized future belief), we utilize two complementary distance metrics:

1. $\ell_1$ Norm (Total Variation Distance): Measures the absolute difference in probability mass assigned to the options.

2. Kullback-Leibler (KL) Divergence: Captures the information-theoretic discrepancy, penalizing cases where the model assigns low probability to the eventual "true" belief.

**Generalization Across Architectures**    The results, visualized in Appendix E, corroborate the findings presented in the main text. We observe that both the two models exhibit a characteristic **burn-in** phase under PPR, characterized by high initial values of both $\ell_1$ distance and KL divergence. However, applying "seeding" strategy or our proposed Martingale Amortization (AM-PPR) significantly suppresses this drift. Notably, AM-PPR consistently achieves the lowest divergence values across both metrics, effectively rendering the belief process self-consistent from the very first token. This confirms that the benefits of amortization are not specific to a single architecture but rather address a fundamental transient instability in autoregressive generation. Furthermore, we validate the scalability of this property of seed marginalization by replicating the identical experiment on OLMO-3.1-32B (Olmo Team et al., 2025). As demonstrated in our expanded results, the exact same qualitative benefit holds at a larger scale, underscoring that our stabilization framework robustly scales up to larger language models.

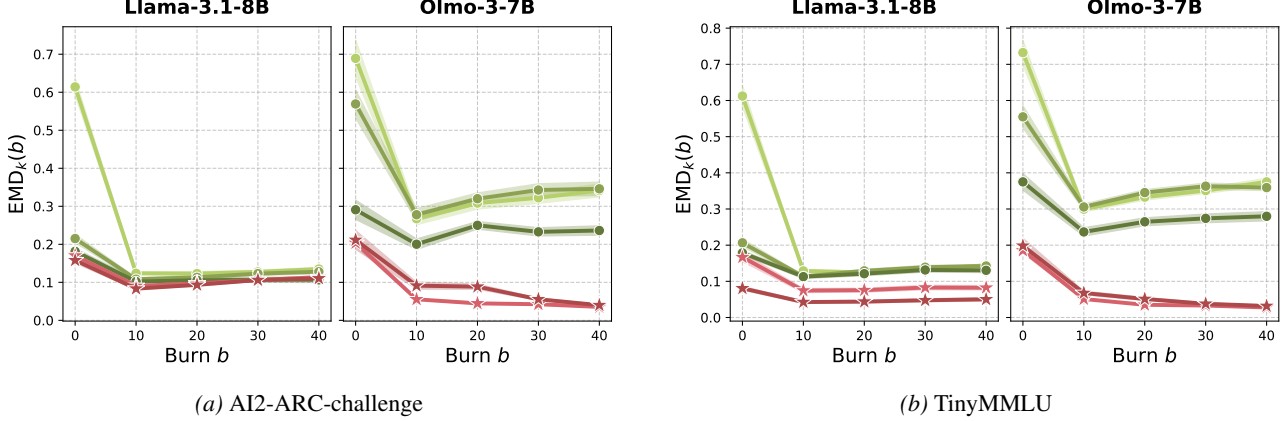

*(a)* AI2-ARC-challenge                                                *(b)* TinyMMLU

*Figure 6.* **Martingale Property Violation Diagnostics by burn-in steps.** We used L1 distance norm in $\mathrm{EMD}_k(b)$ to evaluate how fast the stabilization was achieved in the benchmarks.

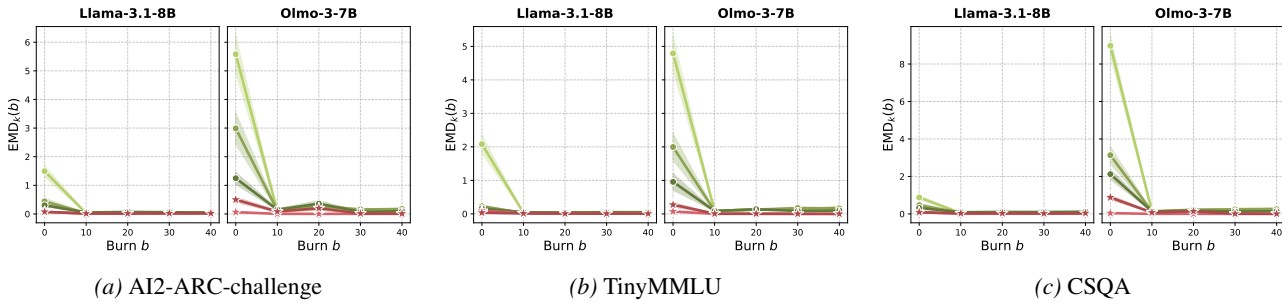

*(a)* AI2-ARC-challenge                    *(b)* TinyMMLU                    *(c)* CSQA

*Figure 7.* **Martingale Property Violation Diagnostics by burn-in steps.** We used KL divergence as metric in $\mathrm{EMD}_k(b)$ to evaluate how fast the stabilization was achieved in the benchmarks.

*Table 5.* Accuracy and AUC score on TinyMMLU, AI2-ARC, CSQA inferred by OLMO-3-7B

| Method | TinyMMLU | | AI2-ARC | | CSQA | |
|---|---|---|---|---|---|---|
| | ACC | AUC | ACC | AUC | ACC | AUC |
| PPR + NoSEED | 44.0 | 71.8 | 63.8 | 85.1 | 57.4 | 86.3 |
| PPR + SEED$_1$ | 49.0 | 72.4 | 67.3 | 86.2 | 60.0 | 87.8 |
| PPR + SEED$_5$ | **54.0** | 75.0 | **74.4** | 89.9 | 73.0 | 86.3 |
| AM-PPR + SEED$_1$ | 51.0 | 77.6 | 74.0 | 89.3 | 74.0 | 88.4 |
| AM-PPR + SEED$_5$ | 52.0 | **79.2** | 73.0 | **91.8** | **77.0** | 92.7 |
| Direct-Query | 52.2 | **82.0** | **74.5** | **92.9** | 75.0 | **94.4** |

*Table 6.* ECE, NLL and Brier Score on TinyMMLU, AI2-ARC, CSQA inferred by OLMO-3-7B

| Method | TinyMMLU | | | AI2-ARC | | | CSQA | | |
|---|---|---|---|---|---|---|---|---|---|
| | ECE | NLL | BS | ECE | NLL | BS | ECE | NLL | BS |
| PPR + NoSEED | **0.0937** | 1.2443 | 0.6734 | 0.1988 | 1.0476 | 0.5542 | 0.2286 | 1.2568 | 0.8625 |
| PPR + SEED$_1$ | 0.1530 | 1.2306 | 0.6670 | 0.2243 | 1.0318 | 0.5427 | 0.2595 | 1.2458 | 0.8775 |
| PPR + SEED$_5$ | 0.1506 | **1.1190** | **0.6458** | **0.1482** | **0.8277** | **0.4160** | 0.2355 | 1.2568 | 0.8625 |
| AM-PPR + SEED$_1$ | **0.1402** | **1.1908** | **0.6386** | 0.1532 | **0.7995** | **0.3916** | **0.1506** | 1.1665 | 0.4234 |
| AM-PPR+ Steer$_5$ | 0.2994 | 1.6778 | 0.7178 | 0.1921 | 1.3286 | 0.4212 | **0.1587** | 1.7489 | **0.3889** |
| Direct-Query | 0.3348 | 1.575 | 0.6925 | 0.1942 | 1.1846 | 0.4172 | 0.1836 | **1.1512** | 0.9439 |

*Table 7.* Accuracy and AUC score on AI2-ARC, CSQA inferred by OLMO-3.1-32B

| Method | AI2-ARC | | CSQA | |
|---|---|---|---|---|
| | ACC | AUC | ACC | AUC |
| PPR + NoSEED | 0.6316 | 0.8384 | 0.5600 | 0.8304 |
| PPR + SEED$_1$ | **0.7520** | **0.8831** | **0.7220** | **0.8956** |
| PPR + SEED$_5$ | **0.7600** | **0.9141** | **0.7520** | **0.9255** |

*Table 8.* ECE, NLL and Brier Score on AI2-ARC, CSQA inferred by OLMO-3.1-32B

| Method | AI2-ARC | | | CSQA | | |
|---|---|---|---|---|---|---|
| | ECE | NLL | BS | ECE | NLL | BS |
| PPR + NoSEED | 0.2241 | 4.6113 | 0.5657 | 0.2812 | 5.2836 | 0.6659 |
| PPR + SEED$_1$ | **0.1750** | **1.1509** | 0.4424 | **0.1822** | **1.2860** | 0.4826 |
| PPR + SEED$_5$ | **0.1455** | 1.6914 | **0.3834** | **0.1125** | 1.3753 | **0.3947** |

# F. Extension to Richer Reasoning Trajectories

The primary experiments focus on the direct-answer MCQA setting where the predictive belief can be explicitly represented and exactly measured on a finite answer simplex. Since core object of our framework is a sequential predictive process, extending this perspective to richer, open-ended reasoning settings is very natural. To provide exploratory empirical evidence supporting our future work proposal, we extend our formulation to complex reasoning paradigms through the lens of In-Context Learning (ICL) in this section.

**Problem Formulation**  For example, in a Chain-of-Thought (CoT) generation setting, our discrete formulation setting can be extended to a continuous setting by projecting open-ended reasoning paths into a continuous embedding space $\mathcal{H}$. Let $R_n = (t_n, a_n)$ denote the $n$-th sequentially generated thought–answer pair with its embedding representation denoted as $H_n = \psi(R_n) \in \mathcal{H}$. In this open-ended regime, we extend our primary object of interest to a history-dependent probability measure over the representation space:

$$\mu_n(\mathbf{x}_Q) := p_\phi(H_{n+1} = \cdot \mid H_{1:n}, \mathbf{x}_Q) \in \mathcal{P}(\mathcal{H}). \tag{47}$$

As $n$ increases, $\mu_n$ traces a stochastic trajectory driven by the model's own generated reasoning paths. Formally, the approximate martingale property translates to verifying whether the future representational belief remains conditionally stable for an appropriate class of bounded test functions $f$ with respect to the filtration $\mathcal{F}_n = \sigma(\mathbf{x}_Q, H_{1:n})$:

$$\mathbb{E}\left[\int f \, d\mu_{n+k} \,\Big|\, \mathcal{F}_n, \mathbf{x}_Q\right] \approx \int f \, d\mu_n. \tag{48}$$

To modulate this continuous predictive process, we naturally extend the direct-query seeding strategy to the continuous domain by modifying the initial context $\mathbf{x}_Q$. Our seeding strategy injects a dynamic history of randomly sampled thought–answer pairs, $S_{1:m} \overset{\text{iid}}{\sim} p_\phi(\cdot \mid Q)$. Crucially, while this initial generation process is essentially identical to standard self-consistency sampling (Wang et al., 2022), our framework uniquely repurposes these samples as an explicit seeding context within $\mathbf{x}_Q$. Rather than serving as static task demonstrations, these randomized pairs leverage the In-Context Learning (ICL) mechanism to modulate the autoregressive context, effectively conditioning and shifting the model's internal activation states within the continuous embedding space $\mathcal{H}$.

**Martingale Consistency**  To evaluate this exploratory proposal and anchor our directions for future work, we conducted a set of preliminary experiments using the larger OLMO-3.1-32B model on complex reasoning benchmarks, namely GSM8K (Cobbe et al., 2021) and GPQA (Rein et al., 2023). The PPR prompts used in this experiment can be found in Figure 9 and Figure 10. To handle the non-discrete nature of the embedding space $\mathcal{H}$, we tracked the alignment of the sequential centroid vectors $\bar{h}_n(\mathbf{x}_Q) := \mathbb{E}_{H\sim\mu_n}[H_n \mid \mathbf{x}_Q]$ of the reasoning paths, which serves as an empirical proxy to diagnose martingale-style consistency. Practically, $\bar{h}_n$ is computed as the empirical mean of the hidden representations across multiple independent rollouts for each query.

Mirroring the qualitative insights from our MCQA experiments, Figure 8 shows that the trajectories exhibit substantial representational drift in the initial steps but becomes increasingly consistent as the sample length grew. Furthermore, when applying PPR with our seeding strategy, we observed significantly faster convergence of this consistency pattern. These results indicate that similar transient belief instabilities arise in more complex scenarios with larger model scales, and designing an appropriate amortized training objective to stabilize the stochastic thinking paths of language models offers a highly promising avenue for future research.

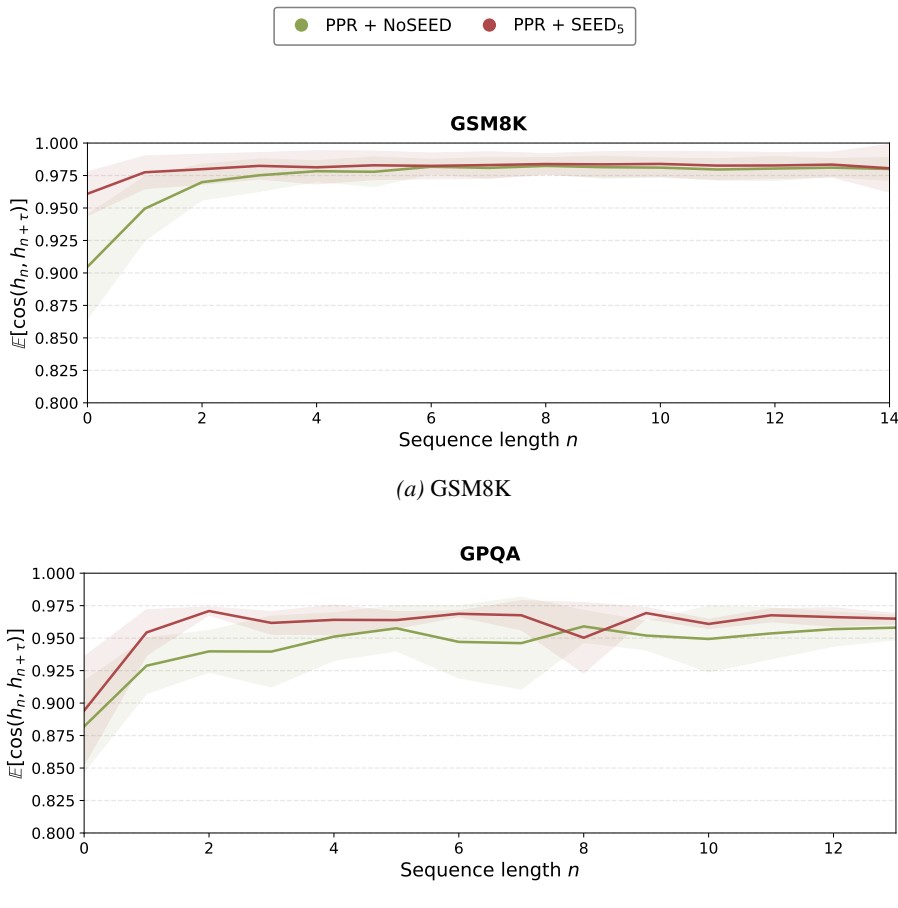

*(a)* GSM8K

*(b)* GPQA

*Figure 8.* **Analysis of martingale property violation in prompted predictive resampling applied to reasoning paths.** We instructed OLMO-3.1-32B to generate a sequence of thought–answer pairs on questions from GSM8K and GPQA. Solid lines indicate the mean cosine similarity between the sequential centroids $\bar{h}_n$ and $\bar{h}_{n+\tau}$ ($\tau = 5$) over 50 questions from each dataset. The shaded region indicates $\pm 1$ standard deviation over the questions. A lower similarity in the initial steps directly captures the representational drift, indicating a transient violation of the martingale property, while reasoning seeding accelerate the stabilization of marginal distributions.

**(GSM8K) PPR Prompt**

```
'''
You are an expert in solving math problems.

## Task
Below is a math problem.
Generate {num_lines} **independent** pairs of (solution, answer).

## Hard Constraint
- Provide **reasonable** solution and answer pairs for each lines **independently**.
- You must NOT cheat on the previous solutions or answers which you have already generated.
- You are NOT allowed to duplicate or omit thinking lines by "Same solution." or "Same answer." of the previous
solutions or answers.

## Output format
Each pair:
<sol>...</sol>
<ans>INTEGER</ans>
Separate pairs with blank lines. No extra commentary outside pairs.
You must generate total {num_lines} pairs.

Question:
'''
```

*Figure 9.* (**GSM8K**) Predictive-Resampling Prompt used for GSM8K. Here, we design a prompt for OLMO-3.1-32B to generate pairs of thought–answer for given GSM8K question.

**(GPQA) PPR Prompt**

```
'''
You are an expert in solving graduate level science problems.

## Task
Below is a graduate level science problem with multiple-choice answers (A˜D).
Generate {num_lines} **independent** pairs of (solution, answer).

## Hard Constraint
- Provide **reasonable** solution and answer pairs for each line **independently**.
- You must NOT cheat on the previous solutions or answers which you have already generated.
- You are NOT allowed to duplicate or omit thinking lines by "Same solution." or "Same answer." of the previous
solutions or answers.

## Output format
Each pair:
<sol>...</sol>
<ans>...</ans>
Separate pairs with blank lines. No extra commentary outside pairs.
You must generate total {num_lines} pairs.

Question:
'''
```

*Figure 10.* (**GPQA**) Predictive-Resampling Prompt used for GPQA. Here, we design a prompt for OLMO-3.1-32B to generate pairs of thought–answer for given GPQA question.

