# OpenReview forum: "From Drift to Coherence: Stabilizing Beliefs in LLMs"
_ICML.cc/2026/Conference — ICML 2026 regular_

### Official Review · Reviewer_3PT1 · 2026-03-12

**Soundness:** 3
**Presentation:** 3
**Significance:** 1
**Originality:** 3
**Overall Recommendation:** 3
**Confidence:** 1

**Summary:**

This paper studies whether an LLM’s predictive beliefs remain stable and coherent when it is asked to answer the same question repeatedly. The authors propose PPR to explicitly expose belief drift, show that the model exhibits early-stage drift but later self-stabilizes, and further introduce seed answers and a self-consistency loss to accelerate or amortize this stabilization process.

**Compliance With Llm Reviewing Policy:**

Affirmed.

**Final Justification:**

I appreciate the author's rebuttal. I currently lack sufficient background knowledge to offer further opinions, so I have set my confidence level to 1 and left the decision-making power to AC.

**Key Questions For Authors:**

Can your method be naturally extended to reasoning settings with chain-of-thought, rather than only direct final-answer prediction? If not, how should readers interpret the applicability of your conclusions to the way modern frontier LLMs are typically used?

**Limitations:**

yes

**Strengths And Weaknesses:**

Strengths

The paper brings the martingale / belief-stability question from a fairly abstract discussion into the concrete setting of LLM prediction dynamics. In addition, the paper provides a reasonably complete combination of theoretical analysis and empirical validation, making the overall argument fairly well supported.

Weaknesses

A main weakness is that the problem setting is still quite toy-like: the experiments are limited to multiple-choice QA, and focus on direct answer generation. This may limit the practical relevance of the work, since frontier models are typically used with chain-of-thought or other extended reasoning processes, which are not considered here.

---

> ### Author Rebuttal · Authors · 2026-03-31
>
> **[W1, Q1] Extension from direct-answer MCQA to reasoning / chain-of-thought settings** \
>
> Thank you for this important and helpful comment. As you rightly understood, our framework is centered on the evolution of a model’s **predictive belief state** under repeated resampling, and on asking whether this sequential predictive process is approximately martingale/coherent. In the current paper, we instantiate this framework in a direct-answer MCQA setting because the predictive belief can be represented explicitly on a finite answer simplex, which makes belief drift and martingale violation directly measurable at the distribution level rather than only indirectly through sampled outputs.
> At the same time, the framework itself is not inherently tied to direct-answer MCQA. Its core object is a sequential predictive process, and extending this perspective to richer reasoning settings is very natural. For example, in a chain-of-thought setting, one can treat each sample as a generated reasoning–answer pair $(s_1,a_1),\ldots,(s_n,a_n)$, and let $\psi(s,a)\in\mathcal H$ denote an embedding or representation of each reasoning–answer pair in a suitable space $\mathcal H$. One can then study a context-dependent predictive process on the induced probability measures $\mu_n \in \mathcal P(\mathcal H)$, rather than on the finite answer simplex. At a high level, the martingale question remains the same: whether, for an appropriate class of bounded test functions $f$, the future representational belief remains conditionally stable,
> \[
> \mathbb E\!\left[\int f\, d\mu_{n+k}\mid \mathcal F_n,Q\right]\approx \int f\, d\mu_n.
> \]
> This is conceptually aligned with our current theory, which already formulates approximate martingale behavior through bounded test functions and conditional predictive stability.
>
>
> So our view is not that the framework is confined to direct final-answer prediction, but rather that the present paper studies it in a regime where the predictive object is especially transparent and exact diagnostics are available. Extending the same martingale/coherence perspective to richer reasoning trajectories, together with practical diagnostics and training objectives for such outputs, is therefore a very natural and important next step. We will clarify this point more explicitly in the revision.
> Also, as a proof of concept, we conducted a small set of additional experiments to examine whether the same qualitative phenomenon extends beyond direct final-answer prediction. Here, we used GPQA and GSM8K using OLMo-3.1-32B to generate a length-$N$ sequence of solution-answer pairs. We extended our probability measure of interest over reasoning trajectories using the embedding space. Specifically, each solution-answer component is autoregressively sampled from the representation space corresponding to the given query. As evidence of martingale-style consistency, we observed that the centroid vectors($\bar{s}_i$) of the solution paths became increasingly consistent as the sample length grew, while exhibiting substantial drift in the initial steps. Furthermore, when applying PPR with seeding, we observed faster convergence of this consistency pattern, which is aligned with the results reported in the main paper. These results indicate that similar phenomena arise in the more complex scenario with the larger model. Designing an appropriate training objective for such richer reasoning outputs is an important direction for future work. With the reproduction of the experiments on the MCQA query set using OLMo-3.1-32B, the experimental results can be found at the following link:
> https://anonymous.4open.science/r/ppr2026-8FFC/_ICML2026__Martingale_LLM__Rebuttal___Copy_.pdf
> You may refer to Figure 1 for the consistency plots on GPQA and GSM8K, and to Table 1 for the reproduced MCQA results.
> We will attach these additional results to the final manuscript.

---

> > ### Author Rebuttal · Reviewer_3PT1 · 2026-04-04
> >
> > Thanks to the author for the rebuttal.

---

> > > ### Author Response · Authors · 2026-04-04
> > >
> > > Thank you very much for your thoughtful and constructive feedback. We sincerely appreciate your insightful comments, which not only suggest promising directions for future work but also help us improve the paper by offering more convincing rationale. We will carefully reflect your feedback in the revision and improve the paper accordingly.
> > >
> > >  We would also be truly grateful if you would consider raising your score in light of the fact that your concerns have been resolved.

---

### Official Review · Reviewer_nqTn · 2026-03-13

**Soundness:** 2
**Presentation:** 3
**Significance:** 2
**Originality:** 2
**Overall Recommendation:** 4
**Confidence:** 3

**Summary:**

This paper studies a transient belief drift in practical LLMs and shows that it can be mitigated. By leveraging the discrete answer space, the authors compute exact predictive distributions and analyze the belief dynamics induced by autoregressive answer resampling. They propose prompted predictive resampling (PPR), in which the model generates a sequence of answers to the same question. Empirically, the results suggest that the resulting stabilization of predictive distributions can improve uncertainty calibration while largely preserving accuracy on several benchmarks.

**Compliance With Llm Reviewing Policy:**

Affirmed.

**Final Justification:**

My biggest concern (Relation between internal coherence and correctness) is addressed.

**Key Questions For Authors:**

Besides the questions in Weakness,  it would be better to investigate the relation between belief and correctness with more detailed empirical evaluations or theoretical analysis. If my biggest concern is well addressed, I tend to raise my score.

**Limitations:**

yes

**Strengths And Weaknesses:**

Strength

1. This paper is well-written and easy to follow.

2. The problem formulation is clear and  well motivated.

3. There is theoretical guarantees for empirical claim.

Weakness

1. My biggest concern is that the internal coherence is not correctness w.r.t. the ground truth. This is the main limitation, since stable predictive process does not mean that the models become more accurate.

2. The method appears sensitive to prompt design and model calibration. From my understanding,  a stable model without being calibrated cannot achieve accurate inference.

3. The theoretical results rely on strong consistency assumptions on MC.

4. In experimental sect., accuracy (table 1) may not be the appropriate metric, since improved stability does not imply improved correctness.

---

> ### Author Rebuttal · Authors · 2026-03-31
>
> **[W1, W4, Q1] Relation between internal coherence and correctness**
>
> Thank you for raising a very important and central question about our method. As you pointed out, improved internal coherence does **not** by itself imply improved correctness with respect to the ground truth. We fully agree with this distinction, and our paper does not intend to claim that martingale consistency is sufficient for higher accuracy.
>
> Our method is designed to reduce martingale violation in the model’s own predictive process, not to directly optimize ground-truth accuracy. In this sense, the paper focuses on internal predictive consistency rather than external likelihood recovery. Accordingly, accuracy is included only as a secondary no-regression check, that is, to verify that improved stability does not substantially harm task performance, rather than as the primary metric for the coherence claim.
>
> Our empirical claim is more limited. Repeated predictive resampling changes the conditioning regime: instead of answering a query under the model’s usual direct-query setting, the model is repeatedly conditioned on its own previously generated answers. Our main observation is that this induces an early-stage belief drift before the process stabilizes. In this regime, the short-horizon predictive belief can differ from the model’s original direct-query belief, and the resulting answer distribution can therefore differ from the one obtained under standard querying.
>
> This is precisely why i.i.d. seeding matters. The seed answers are sampled independently from the model’s standard direct-query distribution $p_\phi(\cdot\mid Q)$, and then prepended as context. Intuitively, these seeds provide an initial history already aligned with the model’s original predictive behavior, which reduces the early burn-in and helps the sequential process start in a regime closer to the direct-query distribution. In this sense, seeding is not introducing a new source of correctness; rather, it mitigates the transient distortion caused by sequential self-conditioning and helps recover behavior closer to the model’s original i.i.d.-style regime.
>
> While we do not claim that improved coherence is sufficient for correctness, our interpretation is that reducing early-stage belief distortion helps preserve the model’s original direct-query behavior, which is why this phenomenon can still matter for downstream predictive correctness. We will revise the paper to clearly note this insightful perspective to make our paper more readable: coherence is a property of internal predictive consistency, not a guarantee of correctness, and martingale consistency should be understood as a necessary, not sufficient, condition for correct or well-calibrated prediction.
>
> **[W2] Sensitivity to prompt design and calibration**
>
> Thank you for the comment. As we discussed in the conclusion, we agree that prompt design can influence the behavior of the predictive process, and improved stability by itself should not be interpreted as guaranteeing full calibration or accurate inference. However, our main claim is that the martingale perspective provides a useful framework for characterizing and measuring **internal predictive drift** in language models, and our contribution is to show that this drift can be substantially reduced through prompted predictive resampling, seeding, and amortization. In this sense, our work identifies and mitigates one concrete source of inconsistency in the model’s sequential predictive behavior.
>
> **[W3] Strength of the theoretical assumptions regarding Monte Carlo consistency**
>
> Thank you for the thoughtful comment regarding our theoretical results. We would like to clarify that the MC-related assumption is mainly a standard estimation-accuracy requirement rather than a restrictive modeling assumption. In our theorem, $\hat p_{\phi,n}^{(k)}$ is obtained by averaging rollout-based predictive estimates, so (A1) only requires that this empirical estimator becomes accurate as the number of rollouts increases. Since the estimator is formed from bounded probability vectors and the implementation uses only a finite truncated set of horizons, this is a feasible and standard large-sample assumption in our setting.
>
> More importantly, the theorem is not meant to state that optimization automatically yields a martingale process under arbitrary conditions; rather, it explains why the SC objective is the right surrogate, in the sense that small horizon-wise discrepancy implies a small martingale residual. In this sense, the assumptions do not introduce a separate unrealistic oracle condition, but rather formalize the regime in which the training objective has succeeded. We will revise the paper to make this interpretation clearer and to avoid giving the impression that we are claiming an unconditional convergence theorem.

---

> > ### Author Rebuttal · Reviewer_nqTn · 2026-04-04
> >
> > Thanks for ur response. I decided to raise my score

---

> > > ### Author Response · Authors · 2026-04-04
> > >
> > > Thank you very much for your thoughtful and constructive comments. We especially appreciate your questions on the core aspects of our paper. They helped us recognize where additional clarification and deeper explanation would most benefit the reader. We will carefully reflect your feedback in the final revision to improve the paper’s clarity and strengthen the reader’s understanding of our main contributions.

---

### Official Review · Reviewer_RQ1p · 2026-03-13

**Soundness:** 2
**Presentation:** 3
**Significance:** 2
**Originality:** 3
**Overall Recommendation:** 4
**Confidence:** 3

**Summary:**

This paper studies LLM uncertainty in multiple-choice QA by prompting the model to answer the same question repeatedly in a single generation, observing that the model's answer distribution shifts substantially in the first few steps before stabilizing. They propose two fixes: (1) prepending a few sampled answers as "seeds" before the sequential generation, and (2) fine-tuning with a loss that forces the model's early-step answer distribution to match its later-step distribution. Experiments on CSQA, AI2-ARC, and TinyMMLU with Llama-3.1-8B and OLMo-3-7B show that the proposed method improves calibration without hurting downstream performance.

**Compliance With Llm Reviewing Policy:**

Affirmed.

**Key Questions For Authors:**

* Do we observe similar pattern for models with bigger sizes (14/30/70B)?
* How does the proposed method compared to other prompting methods, such as self-consistency sampling?
* How to extend the proposed framework to more realistic tasks beyond multiple choice questions?

**Limitations:**

Yes

**Strengths And Weaknesses:**

**Strength**

* LLM calibration is a well-motivated problem.
* The proposed method is simple and intuitive.
* The paper is clearly written.

**Weakness**
* The experiments and methods focus on multiple-choice QA questions, which are pretty rare in real-world uses of LLMs.
* Please refer to the below section for further questions regarding experiment settings.

---

> ### Author Rebuttal · Authors · 2026-03-31
>
> **[W1, Q1, Q3] Extension to more realistic tasks beyond multiple-choice QA using larger models** \
> Thank you for this important and helpful comment. As you rightly understood, our framework is centered on the evolution of a model’s **predictive belief state** under repeated resampling, and on asking whether this sequential predictive process is approximately martingale/coherent. In the current paper, we instantiate this framework in a direct-answer MCQA setting because the predictive belief can be represented explicitly on a finite answer simplex, which makes belief drift and martingale violation directly measurable at the distribution level rather than only indirectly through sampled outputs.
> At the same time, the framework itself is not inherently tied to direct-answer MCQA. Its core object is a sequential predictive process, and extending this perspective to richer reasoning settings is very natural. For example, in a chain-of-thought setting, one can treat each sample as a generated reasoning–answer pair $(s_1,a_1),\ldots,(s_n,a_n)$, and let $\psi(s,a)\in\mathcal H$ denote an embedding or representation of each reasoning–answer pair in a suitable space $\mathcal H$. One can then study a context-dependent predictive process on the induced probability measures $\mu_n \in \mathcal P(\mathcal H)$, rather than on the finite answer simplex. At a high level, the martingale question remains the same: whether, for an appropriate class of bounded test functions $f$, the future representational belief remains conditionally stable,
> \[
> \mathbb E\!\left[\int f\, d\mu_{n+k}\mid \mathcal F_n,Q\right]\approx \int f\, d\mu_n.
> \]
> This is conceptually aligned with our current theory, which already formulates approximate martingale behavior through bounded test functions and conditional predictive stability.
>
>
> So our view is not that the framework is confined to direct final-answer prediction, but rather that the present paper studies it in a regime where the predictive object is especially transparent and exact diagnostics are available. Extending the same martingale/coherence perspective to richer reasoning trajectories, together with practical diagnostics and training objectives for such outputs, is therefore a very natural and important next step. We will clarify this point more explicitly in the revision.
> Also, as a proof of concept, we conducted a small set of additional experiments to examine whether the same qualitative phenomenon extends beyond direct final-answer prediction. Here, we used GPQA and GSM8K using OLMo-3.1-32B to generate a length-$N$ sequence of solution-answer pairs. We extended our probability measure of interest over reasoning trajectories using the embedding space. Specifically, each solution-answer component is autoregressively sampled from the representation space corresponding to the given query. As evidence of martingale-style consistency, we observed that the centroid vectors($\bar{s}_i$) of the solution paths became increasingly consistent as the sample length grew, while exhibiting substantial drift in the initial steps. Furthermore, when applying PPR with seeding, we observed faster convergence of this consistency pattern, which is aligned with the results reported in the main paper. These results indicate that similar phenomena arise in the more complex scenario with the larger model. Designing an appropriate training objective for such richer reasoning outputs is an important direction for future work. With the reproduction of the experiments on the MCQA query set using OLMo-3.1-32B, the experimental results can be found at the following link:
> https://anonymous.4open.science/r/ppr2026-8FFC/_ICML2026__Martingale_LLM__Rebuttal___Copy_.pdf
> You may refer to Figure 1 for the consistency plots on GPQA and GSM8K, and to Table 1 for the reproduced MCQA results.
> We will attach these additional results to the final manuscript.
>
>
>
> **[Q2] Comparison to other prompting methods**
>
> Thank you for this insightful question. We agree that comparison with other prompting methods, such as self-consistency, is important for clarifying the practical meaning of our method. In fact, the performance reported as **Direct-query** in the main paper should be understood in essentially this spirit: it is measured by aggregating multiple sampled answers and making the final decision by voting, i.e., in a way closely related to standard self-consistency evaluation. We will add a clearer explanation of this point in the final manuscript to avoid confusion.

---

> > ### Author Rebuttal · Reviewer_RQ1p · 2026-03-31
> >
> > Thank you for the response! The new experiments with OLMO-32B on math and reasoning benchmark makes the finding more generalizable. I have increased my score to 4.

---

> > > ### Author Response · Authors · 2026-04-02
> > >
> > > Thank you very much for the thoughtful questions and feedback. We really appreciate it, and we are glad to share experiment results on more scalable model and extended framework. Your comments also helped us identify several directions that we agree would make strong future work. We will make sure to incorporate your feedback carefully in the final revision and improve the paper accordingly.

---

### Official Review · Reviewer_Cyd1 · 2026-03-23

**Soundness:** 3
**Presentation:** 3
**Significance:** 2
**Originality:** 3
**Overall Recommendation:** 4
**Confidence:** 3

**Summary:**

This paper proposes a prompt engineering method to show whether LLM predictive behavior can be seen through the martingale and Bayesian lens. The authors show that  the LLM belief has an initial transient drift at the beginning of a repeated answering sequence but then stabilizes as n ( sequence length) increases, so the belief process can be interpreted as a martingale posterior without that initial drift.
They authors then propose a finetuning strategy(white box), as well as a answer-seeding strategy (blackbox) that both minimize the transient belief drift in the beginning of the answer sequence. Theoretical justification is provided for the self-consistency loss function they optimize, and experimentally they show that their proposed finetuning and answer seeding both effectively reduce the average expected mean drift, as well as how optimizing for martingale property by minimizing the initial drift can yield more accurate calibrated uncertainty estimates.

**Compliance With Llm Reviewing Policy:**

Affirmed.

**Final Justification:**

I will note that my questions are clarified. Given my concerns were mostly clarification questions, I will maintain my score of 4.

**Key Questions For Authors:**

1. In section 4.2, the authors argue that belief process settles into a stable equilibrium as  the sequence grows, and that the model possess a coherent true belief state that it fails to access immediately. But when I look at Figure 1, the distribution that the model stabilizes to with answer seeding seems visually different from the distribution it stabilizes to without answer seeding. If there is a single underlying true belief state why do these two examples ( with and without answer seeding) converge to different distributions? could the authors comment on whether the stabilized distribution is genuinely unique or whether it depends on the context provided, for example the sequence of the answer seeds.


2. with respect to how the answer seeding is implemented: when generating the independent trajectories, is the same set of seed answers used as context for all of the random draws of the trajectories or is it a fresh set of seed answers drawn independently for each trajectory instantiation? This matters because I suspect that different seed draws lead to different stabilizing distributions as seen in the LLama3 Figure 1. Could the authors clarify this and comment on how sensitive the stabilization distribution is to the specific seed answers ?


3. could you comment more on the three assumptions in line 307 - 313. Specifically Assumption and the practical conditions under which these assumptions hold?

**Limitations:**

Yes

**Strengths And Weaknesses:**

Strengths:

1. The empirical findings that the LLM belief exhibit an initial transient drift but eventually stabilize to exhibit martingale property is interesting and practically meaningful.

2. The extension of theorem 4.2 to unseen queries is an important contribution since it shows that the behavior learned during training generalized to unseen queries which shows its not just a mere memorization.

3. The experiments are comprehensive and sound, and the evaluation of downstream impact on calibration and accuracy is well-motivated and connects the findings about the theoretical notion of the martingale property to a practical downstream use-case.



Weaknesses:

1. The paper only considers multiple choice questions, and most of experiments and assumptions depend on being able to obtain the next token probabilities ( where in multiple choice its just the probability of the answer choice token so its easier to work with)  over a finite and known answer set; and I understand the motivation for clean theory and observations and the authors do justify this choice in the beginning. However, given that the authors propose a fine-tuning strategy, this restriction significantly limits the practicality of the finetuning method  to broader applications that are more common with LLMs. I believe it would strengthen the paper if the authors added a section, even in the appendix where they discuss how the training strategy could be extended to open-ended generation beyond multiple choice.

2. There were several technical terms and analogies made in the paper but were not clearly or technically explained, which I found vague.  (e.g the MCMC analogy is invoked repeatedly in the introduction without a precise technical statement, an another example is the "mean-field summary" in line 187 is used without definition or citation, and reads as jargon rather than precise technical term. Also "generic question answering" in line 208 is rather misleading given the authors only consider multiple choice questions which is not generic question answering by standard definition ( I would think of free-form response as generic given the LLM application).

3. Please also see the questions below.

---

> ### Author Rebuttal · Authors · 2026-03-31
>
> **[W1] Extension beyond multiple-choice questions / open-ended generation**
>
> Thank you for this important point. We have extended our experiments to a larger model and to thinking-style tasks, specifically GSM8K and GPQA with OLMo-3.1-32B. In this setting, we treat each **solution–answer pair** as an instance and analyze the martingale consistency of the induced predictive process over these pairs, rather than only over final answers.
>
> Empirically, we observed the same qualitative phenomenon as in the main paper: even in these reasoning-style settings with a larger model, the predictive process shows drift in the early stage and becomes stable as the sample length grows, while seeding further accelerates this stabilization. We believe these results support that the core phenomenon is not limited to direct-answer MCQA. At the same time, extending the SC loss itself to such richer reasoning outputs is a natural and promising direction for future work. For the corresponding experimental setup and detailed results, please refer to our response to Reviewer 3PT1, where we provide a fuller description of this extension.
>
> **[W2] Vague terminology and imprecise framing**
>
> Thank you for this helpful comment. We agree that some of the terminology and analogies in the current draft are not precise. In the final manuscript, we will revise these parts to use more technically accurate and reader-friendly wording, including clarifying or removing vague expressions.
>
> **[Q1, Q2] Uniqueness of the stabilized belief/dependence on seed context**
>
> Thank you for raising these important clarification questions. We agree that Figure 1 requires a more precise interpretation. Our intent was not to claim a single context-independent stabilized distribution shared by seeded and unseeded prompting. In our framework, the predictive belief process is defined conditional on the initial context $x_Q$:
>
> $\theta_n(x_Q):=p_\phi(A_{n+1}=\cdot \mid A_{1:n},x_Q)$.
>
> Thus, unseeded prompting uses $x_Q=[I;Q]$, while seeded prompting uses $x_Q=[I;Q;S_{1:m}]$. These, therefore, induce different conditional belief processes, so there is no reason a priori that they must converge to the same stabilized distribution. More precisely, the relevant limits are $\theta_\infty([I;Q])$ and $\theta_\infty([I;Q;S_{1:m}])$, and the latter can depend on the particular seed realization. In this sense, seeding is not only a burn-in accelerator but also a steering intervention: it can reduce early-stage drift while also shifting the eventual stabilized distribution.
>
> Regarding implementation, the seed prefix is not shared across all $J$ rollouts. For each rollout $j$, we draw a fresh seed sequence $S_{1:m}^{(j)}\sim p_\phi(\cdot\mid Q)$, construct $x_Q^{(j)}=[I;Q;S^{(j)}_{1:m}]$, and then run predictive resampling from that trajectory-specific context. Therefore, the seeded estimator averages over both rollout randomness and seed randomness, so it is best interpreted as a seed-marginalized quantity rather than one conditioned on a single fixed seed prefix.
>
> We agree that this distinction matters, since different seed draws can in principle induce different stabilized distributions. In qualitative checks, we also observed that unusually unfavorable seed prefixes can shift the subsequent answer-sequence distribution. However, this is not the regime targeted in our main experiments: the seeds are sampled from the model’s own direct-query distribution rather than chosen adversarially. We will revise the paper to make these points explicit.
>
> **[Q3] Practical meaning of the theoretical assumptions**
>
> Thank you for the question about our theoretical results. We agree that explaining more clearly what the assumptions mean and under what practical conditions they hold would make the paper easier to follow.
>
> (A1) is an **estimation assumption**: the rollout-based estimator $\hat p_{\phi,n}^{(k)}$ should accurately approximate the corresponding horizon-wise predictor. In practice, this requires enough rollouts for stable estimates on the finite horizon set used in training/evaluation. (A2) is a **post-training condition**: after optimization, horizon-wise predictors should be close, which is exactly what the SC loss is designed to enforce. In practice, this means training has reduced the SC objective enough that short- and long-horizon predictions align. (A3) is a **coverage/connectivity condition**: the sampled horizon pairs must connect the relevant horizon range. In practice, this means the training distribution over horizon pairs assigns positive probability to enough pairs so that closeness can propagate across the truncated horizon set used by the implementation.
>
> With this interpretation, the theorem should be read as
>
> $\text{small SC loss}\Rightarrow\text{small horizon-wise discrepancy} \Rightarrow\text{small martingale residual}$.
>
> We will revise the final version to state these roles more explicitly and to clarify the corresponding practical conditions.

---

> > ### Author Rebuttal · Reviewer_Cyd1 · 2026-04-01
> >
> > Thank you for the rebuttal. It clarified most of my questions, and I appreciate the additional explanation.
> >
> > That said, I still remain concerned about the interpretation of the stabilizing distribution claim. In particular, the rebuttal now makes clear that different initial seed realizations can not only accelerate access to a stabilized regime, but can also change the eventual stabilized distribution itself. This seems difficult to reconcile with the paper’s original framing that the method helps the model access an underlying stabilized belief state more quickly. As clarified in the rebuttal, the relevant limiting distribution is context-dependent and, in the seeded case, may vary with the seed realization. This suggests that seeding is not merely reducing burn-in toward a fixed target, but is also steering the process toward a different limiting distribution. I therefore find the original framing somewhat misleading, and I believe the paper should present this distinction more explicitly.

---

> > > ### Author Response · Authors · 2026-04-02
> > >
> > > Thank you for your deep reading of our paper and for continuing this constructive discussion. We agree that the wording in the current manuscript can be read as if seeded PPR were simply a faster route to a single prompt-independent limiting belief, and we understand why this is confusing. We will revise the manuscript to make this distinction more explicit. Our intended point is more nuanced. For a fixed seed realization, the seeded process is indeed conditional on a different initial context, so its limiting distribution need not coincide pathwise with the unseeded one. However, the seeds in our method are not arbitrary external tokens: they are sampled from the model's own direct-query distribution. In this sense, seeded PPR should be viewed not as arbitrary prompt steering, but as an endogenous stochastic initialization anchored to the model's direct-query belief.
> > >
> > > This distinction is important. Because the seeded procedure averages over many independently drawn seed prefixes, the relevant object is not the limit under one particular seed realization, but a **seed-marginalized** stabilized process induced by the model's own direct-query distribution. So our claim is not that every seeded trajectory converges to exactly the same conditional limit as unseeded PPR, but rather that stochastic self-seeding provides a more efficient route to a stabilized regime that remains anchored to the model's original predictive behavior, rather than being determined by arbitrary exogenous context.
> > >
> > > This is also how we interpret the empirical results. Seeding consistently reduces EMD and shortens burn-in, while maintaining competitive downstream performance. So the role of seeding in our framework is not merely to change the target distribution, but to initialize the predictive process with samples drawn from the model's own belief in a way that makes convergence to a stable regime faster and empirically more reliable on average. We will revise the manuscript accordingly and state this more precisely: seeded PPR should be interpreted as stochastic self-seeding toward a seed-marginalized family of context-conditional stabilized beliefs, rather than as access to a single seed-independent limiting distribution. We will make sure to reflect this discussion explicitly in the final manuscript. We again thank you for this thoughtful and constructive discussion.

---

### Decision · Program_Chairs · 2026-04-30

**Decision:**

Accept (regular)

**Comment:**

This paper studies LLM belief drift under repeated answering and proposes methods to improve prediction coherence in multiple-choice QA. On the positive side, reviewers indicate that the paper is well motivated, with meaningful empirical findings and relatively solid experiments. Prior to the rebuttal, the main concerns centered on the limited scope of multiple-choice questions, unexplained terms, and clarification of experimental details. The authors provided a careful rebuttal and resolved most of these concerns. After the rebuttal, most reviewers remained positive. While reviewer Cyd1 raised a remaining concern regarding the interpretation of the stabilizing distribution claim, the authors’ reply clarified this point, and the reviewer explicitly stated that their questions were clarified. Reviewer 3PT1 maintained a negative score, but the reviewer also stated that the concern had been fully resolved and set the confidence to 1, so the AC places less weight on this assessment. Overall, the AC believes the rebuttal resolved the main reviewer concerns and therefore recommends accept.